# Caffeine blocks SREBP2-induced hepatic PCSK9 expression to enhance LDLR-mediated cholesterol clearance

Paul F. Lebeau[1,13], Jae Hyun Byun[1,13], Khrystyna Platko[1], Paul Saliba [2], Matthew Sguazzin[2], Melissa E. MacDonald[1], Guillaume Paré [3,4,5], Gregory R. Steinberg [2,6], Luke J. Janssen[7], Suleiman A. Igdoura[8], Mark A. Tarnopolsky [9,10], S. R. Wayne Chen[11], Nabil G. Seidah[12], Jakob Magolan [2] & Richard C. Austin [1,5 ✉]

Evidence suggests that caffeine (CF) reduces cardiovascular disease (CVD) risk. However, the mechanism by which this occurs has not yet been uncovered. Here, we investigated the effect of CF on the expression of two bona fide regulators of circulating low-density lipoprotein cholesterol (LDLc) levels; the proprotein convertase subtilisin/kexin type 9 (PCSK9) and the low-density lipoprotein receptor (LDLR). Following the observation that CF reduced circulating PCSK9 levels and increased hepatic LDLR expression, additional CF-derived analogs with increased potency for PCSK9 inhibition compared to CF itself were developed. The PCSK9-lowering effect of CF was subsequently confirmed in a cohort of healthy volunteers. Mechanistically, we demonstrate that CF increases hepatic endoplasmic reticulum (ER) $Ca^{2+}$ levels to block transcriptional activation of the sterol regulatory element-binding protein 2 (SREBP2) responsible for the regulation of PCSK9, thereby increasing the expression of the LDLR and clearance of LDLc. Our findings highlight ER $Ca^{2+}$ as a master regulator of cholesterol metabolism and identify a mechanism by which CF may protect against CVD.

[1] Department of Medicine, Division of Nephrology, McMaster University, The Research Institute of St. Joe's Hamilton and the Hamilton Center for Kidney Research, Hamilton, ON L8N 4A6, Canada. [2] Department of Biochemistry and Biomedical Sciences, McMaster University, Hamilton, ON L8S 4L8, Canada. [3] Population Health Research Institute, McMaster University, Hamilton, ON L8L 2X2, Canada. [4] The Departments of Medicine, Epidemiology and Pathology, McMaster University, Hamilton, ON L8L 2X2, Canada. [5] The Thrombosis and Atherosclerosis Research Institute (TaARI), Department of Medicine, David Braley Research Institute, McMaster University, Hamilton L8L 2X2, Canada. [6] Centre for Metabolism, Obesity and Diabetes Research, Department of Medicine, McMaster University, Hamilton, ON L8S 4L8, Canada. [7] Firestone Institute for Respiratory Health, St. Joseph's Hospital, Hamilton, ON L8S 4K1, Canada. [8] Department of Biology and Pathology, McMaster University, Hamilton, ON L8S 4K1, Canada. [9] Department of Medicine/Neurology, McMaster University, Hamilton, ON L8N 3Z5, Canada. [10] Department of Pediatrics, McMaster University, Hamilton, ON L8S 4K1, Canada. [11] Libin Cardiovascular Institute of Alberta, Department of Physiology and Pharmacology, University of Calgary, Calgary, AB T2N 2T9, Canada. [12] Laboratory of Biochemical Neuroendocrinology, Clinical Research Institute of Montreal, affiliated to the University of Montreal, Montreal, QC H2W 1R7, Canada. [13]These authors contributed equally: Paul F. Lebeau, Jae Hyun Byun. ✉email: austinr@mcmaster.ca

ncreased levels of circulating low-density lipoprotein choles-terol (LDLc) are tightly linked to the development of cardio-vascular disease (CVD). Despite the approval of several therapies that lower LDLc, many patients fail to reach their LDL lowering goal due to intolerance, adverse events, or simply the high cost of medications. An important regulator of LDLc is the sterol regulatory element-binding protein 2 (SREBP2), which is an endoplasmic reticulum (ER)-resident transcription factor. SREBP2 is activated by reductions in intracellular cholesterol and loss of ER $Ca^{2+}$, which then triggers translocation to the nucleus and the induction of cholesterol regulatory genes including the proprotein convertase subtilisin/kexin type 9 (PCSK9), the low-density lipoprotein receptor (LDLR), and HMG-CoA reductase (HMGR)[1]. Recent advancements in therapies available for the management of dyslipidemia and CVD have led to the char-acterization of PCSK9 as a hepatocyte-secreted circulating factor capable of enhancing the degradation of cell-surface LDLR[2–5]. By extension, PCSK9 also reduces the ability of metabolically active tissues, such as the liver, to remove excess LDLc from the blood. Based on these seminal discoveries, anti-PCSK9 antibodies are now available to patients at high risk of CVD, yielding an unprecedented 60–70% reduction of LDLc levels[6]. Although efficacious, the high cost and/or need for subcutaneous administration of anti-PCSK9 antibodies poses a limit to their availability to patients worldwide[7]. Such circumstances warrant the need for additional studies examining the molecular mechanisms that modulate the expression and secretion of PCSK9 from hepatocytes in order to develop more cost-effective therapies.

Caffeine (CF) or 1,3,7 trimethylxanthine, is best known as a stimulant alkaloid of the central nervous system found in various plants and is commonly found in coffee and tea. The majority of published literature demonstrates that the average adult habitual caffeine drinker consumes between 400 and 600 mg of CF daily and organizations like Health Canada and the Food and Drug Administration conclude that such doses are not negatively associated with toxicity, cardiovascular effects, bone status, cal-cium imbalance, behavior, the incidence of cancer or effects on male fertility[8]. On the contrary, accumulating evidence now suggests that moderate to high levels of CF (>600 mg), consumed daily in the form of non-alcoholic beverages, are associated with a reduction in CVD risk[8,9]. Although biochemical studies have shown that CF increases intracellular $Ca^{2+}$ levels and induces vasodilation of the vascular endothelium via release of nitric oxide[10,11], a cellular process known to be cardioprotective[12], molecular mechanisms supporting clinical evidence are currently lacking.

In the current study, we demonstrate that clinically relevant concentrations of caffeine suppress SREBP2 transcriptional acti-vation in liver hepatocytes, thereby leading to a reduction of PCSK9 in both mice and humans. Using structure/activity rela-tionships (SAR), we have also generated several xanthine deri-vatives with heightened antagonism against SREBP2 and PCSK9, compared to CF. Overall, these studies characterize the mechanism by which CF impacts the expression of genes well-known to mediate CVD risk.

## Results

### CF blocks PCSK9 expression and secretion in hepatocytes.
To initiate our studies, cultured immortalized hepatocytes known to express and secrete PCSK9[13], including HuH7 and HepG2 cells, as well as primary mouse- and human-hepatocytes (PMH and PHH, respectively), were treated with CF for 24 h and assessed for PCSK9 expression via immunoblots and real-time PCR (Fig. 1A–D). These initial experiments revealed that CF reduced

protein and mRNA transcript levels of PCSK9. CF also attenuated PCSK9 expression resulting from the SERCA pump antagonist and established ER stress-inducing agent, thapsigargin (TG; Fig. 1A, B)[13]. Given that sterol deprivation represents another well-established promoter of SREBP2 activation[13], cells were also treated with CF in the presence and absence of U18666A (U18), a pharmacological agent that depletes intracellular sterols. Like TG-treated cells, CF attenuated U18-induced PCSK9 expression (Fig. 1A; see Supplementary Table 1 in supplemental materials for a list of compounds and mechanisms of action). Importantly, CF also blocked the secretion of PCSK9 from HuH7 and HepG2 cultured hepatocytes, as well as from PMH and PHH (Fig. 1E–G). Control experiments demonstrated that CF did not interfere with the ELISA by measuring levels of recombinant PCSK9 in the presence or absence of CF (Fig. S1). A Coomassie stain of elec-trophoretically resolved media harvested from these cells was also used to confirm that CF was not affecting global protein secretion (Fig. 1H). Next, HepG2 cells were treated with an increasing dose of CF. Results from this experiment demonstrate that PCSK9 elicited a dynamic response to CF from the $10^2$ to the $10^8$ nM range in hepatocytes (Fig. 1I). To determine whether caffeine was affecting PCSK9 expression and secretion in a transcriptional or post-transcriptional manner, HepG2 cells were pre-treated with a transcription blocker (actinomycin D [ActD]) and subsequently exposed to CF. The failure of CF to block PCSK9 mRNA (Fig. 1J) and secreted protein (Fig. 1K) in the presence of ActD suggests that CF exerts its effect on PCSK9 in a transcription-dependent manner. In support of these findings, CF also failed to block the secretion of PCSK9 in cells transfected with a CMV-driven PCSK9 vector (Fig. 1L). These data demonstrate that CF, in the nanomolar range, can cause a significant reduction of PCSK9 expression at the mRNA and protein levels in a variety of cul-tured hepatocyte cell models and that the inhibition likely occurs at the transcriptional level.

### CF blocks SREBP2 activation in hepatocytes.
Our research group has previously demonstrated that ER stress, specifically resulting from the depletion of ER $Ca^{2+}$, promotes the activation of SREBP2 and the expression of PCSK9[13–15]. We, therefore, examined the effect of CF on TG-induced SREBP2 activation. Consistent with previous studies[16], we observed that CF blocked the expression of SREBP2 in PMHs and PHHs, as well as in HepG2 cells (Fig. 2A–C). CF also blocked the expression of a downstream target of SREBP2 transcriptional activity in PMHs, HMGR (Fig. 2A), as well as SREBP1, the isoform known to regulate fatty acid synthesis (Fig. 2D). The effect of CF on the expression of hepatocyte nuclear factor 1a, a liver-expressed transcription factor also known to regulate PCSK9 expression[17], was assessed but did not yield a significant difference in the absence of TG (Fig. S2). SREBP2 activity was then examined at the protein level in HuH7 cells transfected with a plasmid encoding GFP driven by the sterol regulatory element (SRE-GFP; Fig. 2E–G). Consistent with the real-time PCR data, we observed that CF blocked the nuclear/activated isoform of SREBP2 (nSREBP2; ~60 kDa) and the expression of SRE-driven GFP in the presence and absence of TG. GFP expression was also visualized via immunofluorescent staining and quantified using ImageJ Software. Immunofluorescent staining of SREBP2 in cells treated with TG in the presence and absence of CF also demonstrated that CF attenuated the re-localization of SREBP2 from the perinuclear region to the nucleus (Fig. 2H; nuclei con-taining activated SREBP2 are indicated by white arrows). Given the well-established role of SREBP2 in the transcriptional reg-ulation of PCSK9, our data suggest that CF reduces PCSK9 expression and secretion by antagonizing de novo synthesis.

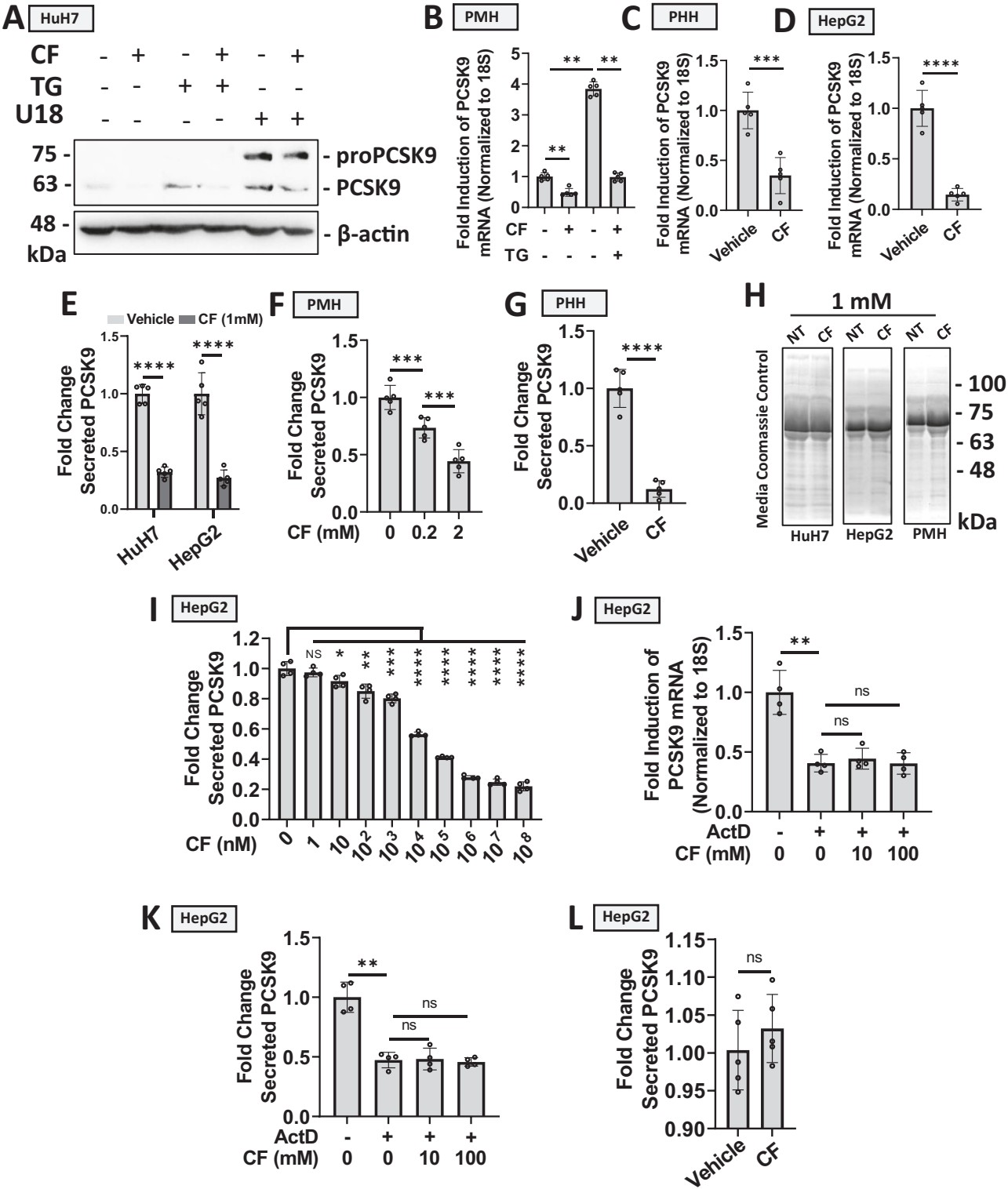

Previous studies have also shown that CF can promote the phosphorylation and activation of AMPK[16,18], a liver-expressed kinase known to induce the inhibitory phosphorylation of SREBP1c[19]. Similar results were observed in our study, whereby CF treatment induced the phosphorylation and activation of AMPK (pAMPK) and subsequent induction of a downstream marker (phosphorylated acetyl CoA carboxylase [pACC]) in the livers of C57BL/6J mice (Fig. S3A). PMHs were then isolated from wildtype (WT) and $Ampk\beta1^{-/-}$ mice[20]. Treatment of these hepatocytes with CF led to a reduction in PCSK9 expression and

secretion (Fig. S3B, C), suggesting that AMPK is not directly involved in CF-mediated PCSK9 inhibition. A similar result was observed in hepatocytes treated with CDN1163 (CDN), a pharmacologic agent known to increase ER $Ca^{2+}$ levels by inducing SERCA pump activation (Fig. S3D, E)[21].

**ER $Ca^{2+}$ modulates PCSK9 expression and secretion.** Among the many intracellular effects of CF, its ability to increase intracellular $Ca^{2+}$ levels is well-studied[11]. Given our previous report

**Fig. 1 Caffeine blocks PCSK9 expression and secretion in hepatocytes. A** HuH7 cells were treated with established inducers of PCSK9 expression, thapsigargin (TG; 100 nM) or U18 (10 μM), in the presence or absence of caffeine (CF; 200 μM) for 24 h[13]. PCSK9 expression was assessed via immunoblot analysis. **B–D** PCSK9 expression was also assessed in primary mouse hepatocytes (PMH) and primary human hepatocytes (PHH), as well as in HepG2 cells treated with CF and TG via real-time PCR ($n = 5$ biologically independent samples per group; data presented are mean ± s.d.). **E–G** PCSK9 ELISAs were assayed on the medium harvested from CF-treated HuH7, HepG2, PMHs, and PHHs ($n = 5$ biologically independent samples per group). **H** Coomassie blue staining of electrophoretically resolved medium harvested from CF-treated cells served to examine the effect of CF on total secreted protein levels. **I** Secreted PCSK9 levels from HepG2 cells treated with an increasing dose of CF ($n = 4$ biologically independent samples per group; data presented are mean ± s.d.). **J-K** PCSK9 expression and secretion were assessed in HepG2 cells treated in the presence and absence of CF and a blocker of transcription, ActD (10 μM) ($n = 4$ biologically independent samples per group; data presented are mean ± s.d). **L** Finally, ELISAs were also used to measure secreted PCSK9 levels in CF-treated HepG2 cells (1 mM) transfected with a CMV-driven PCSK9 vector ($n = 5$ biologically independent samples per group; data presented are mean ± s.d). Statistical comparisons between two groups were conducted using unpaired two-tailed Student's t-tests, while comparisons between multiple groups were compared using one-way ANOVAs with the Tukey HSD post-hoc test (*$p < 0.05$; **$p < 0.01$; ***$p < 0.001$; ****$p < 0.0001$). Source data are provided as a Source Data file.

showing that ER $Ca^{2+}$ depletion induces SREBP2 activation[13], we postulated here that (a) CF may increase ER $Ca^{2+}$ levels, and (b) other agents known to increase ER $Ca^{2+}$ levels may also block SREBP2 activation and PCSK9 expression. To test this hypothesis, we first examined cytosolic $Ca^{2+}$ levels in CF-treated cells using the high-affinity fluorescent $Ca^{2+}$ indicator, Fura-2-AM. Consistent with previous studies, CF significantly increased cytosolic $Ca^{2+}$ levels in immortalized hepatocytes (Fig. S4). ER $Ca^{2+}$ levels were then examined in cells transfected with D1ER; a genetically encoded ER-resident fluorescence resonance energy transfer (FRET)-based calreticulin chameleon $Ca^{2+}$ sensor, which increases in fluorescence intensity upon $Ca^{2+}$ binding[22]. The low-affinity $Ca^{2+}$ indicator, Mag-Fluo-4, was also utilized for the direct assessment of ER $Ca^{2+}$ and increases in fluorescence intensity upon $Ca^{2+}$ binding[23,24]. The fluorescence intensity of cells treated with CF and control agents, TG and CDN, was assessed using a fluorescent spectrophotometer and visualized using a fluorescent microscope (Fig. 3A). In addition to heightened cytosolic $Ca^{2+}$ levels, we also observed that CF increased ER $Ca^{2+}$ levels. As expected, the control agent CDN increased ER $Ca^{2+}$ levels, whereas TG reduced ER $Ca^{2+}$ levels. ER $Ca^{2+}$ content was also assessed indirectly with the high-affinity $Ca^{2+}$ dye, Fura-2-AM (Fig. 3B). HuH7 cells were pretreated with CF for 24 h and subsequently exposed to a high dose of TG, which causes a spontaneous loss of ER $Ca^{2+}$. In response to this, cells pretreated with CF exhibited increased ER $Ca^{2+}$ efflux compared to cells treated with the vehicle control when exposed to TG. We also observed that the protein expression of calnexin, an ER-resident protein with high capacity for $Ca^{2+}$ binding[25], was induced by CF and blocked by TG (Fig. 3C).

To further test our hypothesis that increasing ER $Ca^{2+}$ blocks PCSK9 expression, we treated cells with a variety of well-established $Ca^{2+}$-modulating agents. At low dose (10 nM), ryanodine is known to facilitate ER $Ca^{2+}$ loss by enhancing RyR-mediated $Ca^{2+}$ transients; whereas high dose ryanodine (10 μM) is known to block RyR-mediated ER $Ca^{2+}$ leakage[26]. The compound 2APB also blocks the exit of ER $Ca^{2+}$ by antagonizing IP3Rs[27,28]. In contrast to these two agents that modulate ER $Ca^{2+}$ release, CDN is an established allosteric activator of the SERCA pump and thus increases the entry of $Ca^{2+}$ into the ER[7]. Consistent with our hypothesis, we observed that high-dose ryanodine, 2APB, and CDN, blocked SREBP2 and PCSK9 at the mRNA transcript level in the presence and absence of TG (Fig. 3D– F). These agents also blocked TG-induced expression of the $Ca^{2+}$-dependent chaperone and ER stress marker, GRP78. Consistent with our previous studies, we also observed that ER $Ca^{2+}$ depletion via TG and cyclopiazonic acid (CPA) treatment increased PCSK9 and SREBP2 expression (Fig. 3G, H)[13]. As expected, these established ER stress-inducing agents also increased the expression of GRP78.

Secreted PCSK9 levels in the media harvested from cells treated with $Ca^{2+}$-modulating agents were then assessed using ELISAs. Consistent with real-time PCR findings, we observed that high-dose ryanodine, CDN, and 2APB reduced PCSK9 secretion (Fig. 3I). Overexpression of calnexin and loss-of function ryanodine receptor variants (RyR2$^{E4872A}$ and RyR2$^{A4860G}$), which were previously shown to increase ER $Ca^{2+}$ levels[25,27], also blocked PCSK9 secretion (Fig. 3J, K). In contrast to its effect on PCSK9 mRNA transcript levels, we also observed that TG reduced PCSK9 secretion (Fig. 3L); an observation consistent with our previous study[13]. Sterol deprivation via treatment with U18, which is not known to affect ER $Ca^{2+}$ levels, yielded findings consistent with our previous observations[13] and increased PCSK9 secretion (Fig. 3M). Finally, to confirm that CF blocked PCSK9 secretion in a $Ca^{2+}$-dependent manner, experiments were repeated in HepG2 cells incubated in $Ca^{2+}$-deficient medium for 48 h (Fig. 3N). We previously demonstrated that this treatment caused robust ER stress and likely explains the observed reduction of secreted PCSK9 levels in the absence of CF. Importantly, however, these data reveal that CF failed to antagonize PCSK9 secretion in cells that have been deprived of $Ca^{2+}$. Overall, these data provide strong evidence that ER $Ca^{2+}$ levels not only affect the expression of ER stress markers, but also regulate PCSK9 and SREBP2.

**$Ca^{2+}$ increases the binding capacity of GRP78 for ER-resident SREBP2 and prevents its exit from the ER.** GRP78 is among a number of $Ca^{2+}$-dependent chaperones that play a central role in facilitating a chemical equilibrium that favors elevated $Ca^{2+}$ levels in the ER lumen relative to the cytosol via direct binding/sequestration and buffering[29]. It is estimated that GRP78 increases the $Ca^{2+}$-retaining ability of the ER by 25%[30]. Given that chaperones increase ER $Ca^{2+}$ levels but are also $Ca^{2+}$-dependent in their capacity to bind and fold polypeptides, we investigated whether ER $Ca^{2+}$ could modulate the ability of GRP78 to interact with ER-resident pre-mature SREBP2 (~125 kDa). In support of this notion, previous studies have demonstrated that (a) GRP78 is highly promiscuous in its client specificity[31], capable of binding to one site every 36 amino acids of a randomly generated peptide[32], (b) $Ca^{2+}$ and ATP bind to GRP78 in a cooperative manner and that ATP is necessary for the peptide binding and folding abilities of this chaperone[33], and (c) overexpression of GRP78 can attenuate the activation of SREBPs in response to ER stress[14].

To determine whether increasing ER $Ca^{2+}$ levels enhance GRP78 peptide binding capacity, HuH7 cells were treated with either CDN, which increases ER $Ca^{2+}$ levels, or TG which causes ER $Ca^{2+}$ depletion. Following treatment, the interaction between GRP78 and SREBP2 was examined via immunoprecipitation of the former. By affecting ER $Ca^{2+}$ levels, however, these agents

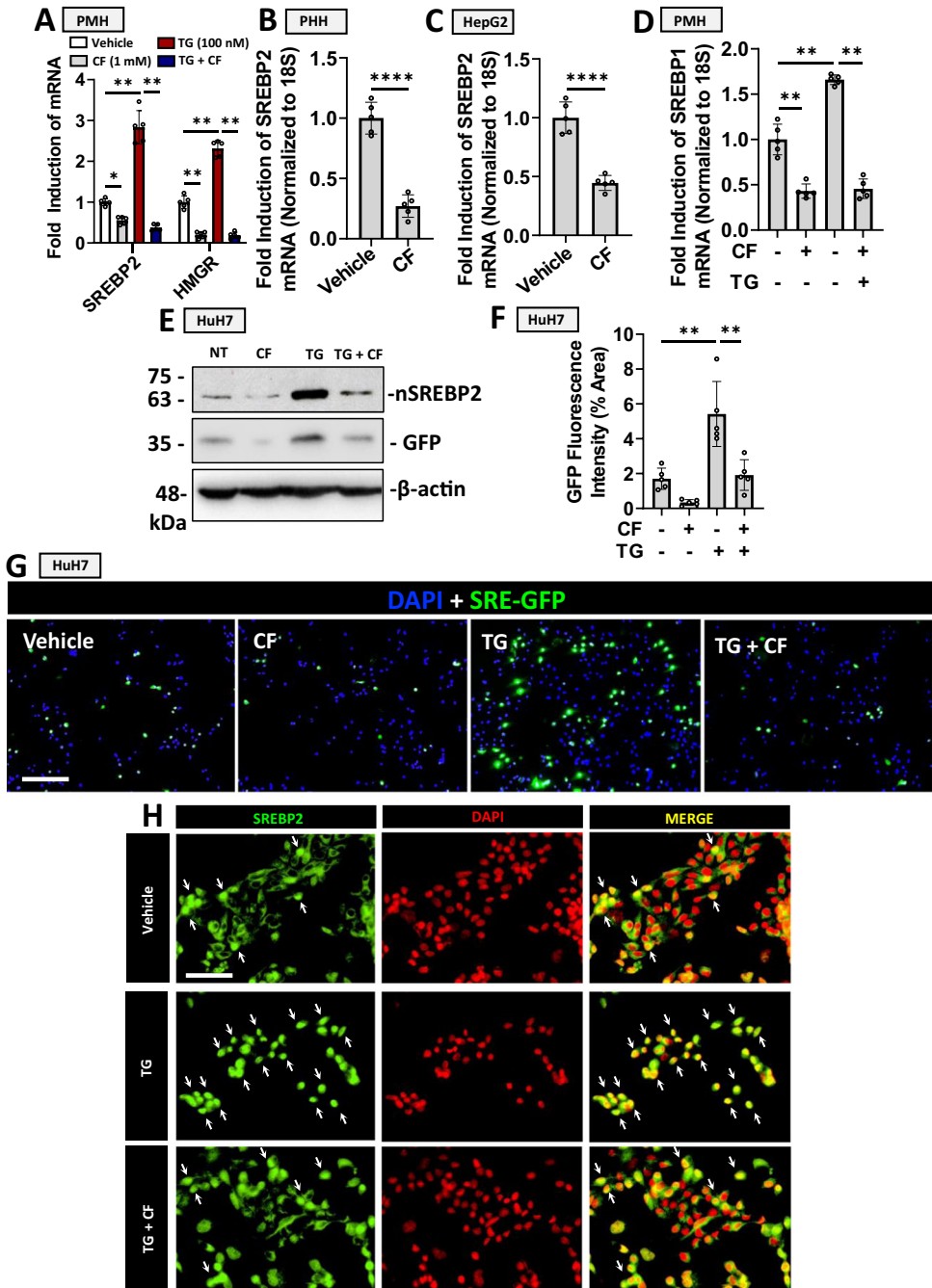

**Fig. 2 Caffeine blocks SREBP2 activation in hepatocytes. A** The effect of caffeine (CF; 200 µM) on SREBP2 and SREBP1 mRNA expression was examined in primary mouse hepatocytes (PMH) in the presence and absence of thapsigargin (TG; 100 nM), an established activator of SREBPs. The downstream product of SREBP2 transcriptional activity, HMGR, was also examined. **B, C** The inhibitory effect of CF on SREBP2 was also examined in primary human hepatocytes (PHH) and HepG2 cells. **D** CF-mediated SREBP1 inhibition was also examined in PMH (*$p < 0.05$). **E–G** HuH7 cells were transfected with a reporter construct encoding a sterol-regulatory element-driven green fluorescent protein (SRE-GFP; green color). Cells were subsequently treated with CF (200 µM) and/or TG (100 nM) 24 h later. GFP and nuclear (n)SREBP2 expression were examined via immunoblot analysis. GFP expression was also assessed via immunofluorescent staining, which was quantified using ImageJ. **H** The cellular localization of SREBP2 (green color) in CF- and TG-treated HuH7 cells was also examined via immunofluorescent staining. Nuclei containing activated SREBP2 are indicated by white arrows. For all data in this figure, $n = 5$ biologically independent samples per group; data presented are mean ± s.d). Scale bars; **G** 100 µm; **H** 20 µm. Statistical comparisons between two groups were conducted using unpaired two-tailed Student's $t$-tests, while comparisons between multiple groups were compared using one-way ANOVAs with the Tukey HSD post-hoc test (*$p < 0.05$; **$p < 0.01$; ***$p < 0.001$; ****$p < 0.0001$). Source data are provided as a Source Data file.

also directly impact the expression and abundance of GRP78 compared to untreated cells. Therefore, assessment of the relative binding capacity GRP78 for SREBP2 between treatments required normalization of immunoprecipitations to equivalent GRP78 protein levels (Fig. 4A). Following conditions of TG-induced ER

$Ca^{2+}$ depletion and stress, GRP78 lost its intrinsic ability to interact with pre-mature SREBP2. Conversely, increasing ER $Ca^{2+}$ levels via CDN treatment enhanced the ability of GRP78 to interact with and sequester pre-mature SREBP2. Strikingly, we observed that CF- and CDN-treated cells behaved similarly,

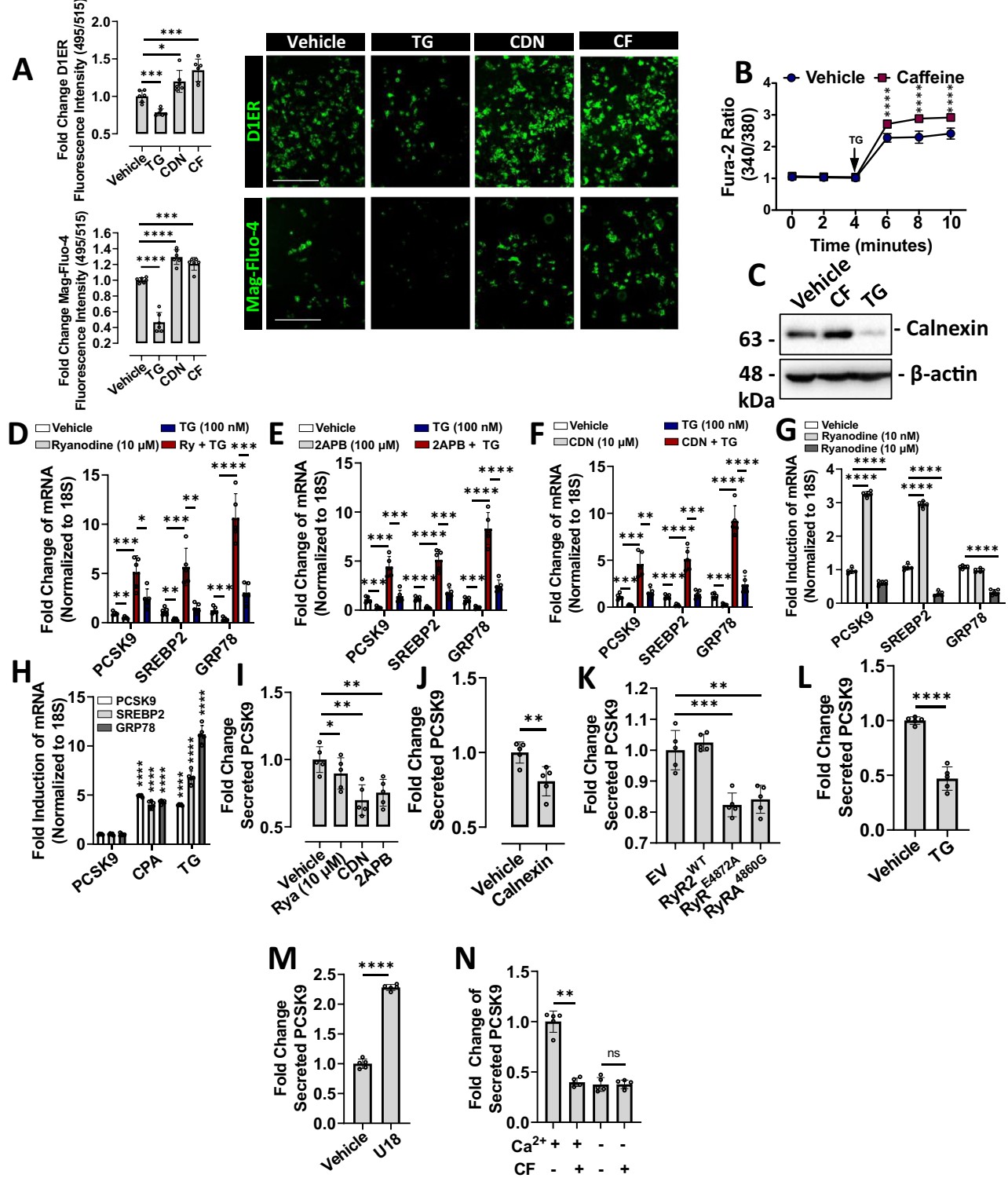

suggesting that CF also increases the SREBP2-binding capacity of GRP78. Consistent with immunoprecipitations, immunoblots of whole-cell lysates demonstrate that CF and CDN antagonized the activation and nuclear localization of SREBP2, whereas TG had the opposite effect (Fig. 4B).

To confirm that CF blocked SREBP2 activation in a manner dependent on GRP78, cells transfected with siRNA targeted against GRP78 (siGRP78) were also treated with CF. Our findings demonstrate that siGRP78 treatment significantly increased the mRNA and secreted forms of PCSK9, as well as the mRNA levels

of SREBP2 (Fig. 4C, D). Consistent with these findings, we also observed that CF failed to attenuate PCSK9 and SREBP2 mRNA expression or secreted PCSK9 levels in the presence of siGRP78 (Fig. 4D, E). The siRNA-mediated knockdown of GRP78 was confirmed via immunoblotting (Fig. 4F).

Because ER $Ca^{2+}$ depletion induces a compensatory unfolded protein response (UPR), we also postulated that CF may attenuate UPR marker expression by increasing ER $Ca^{2+}$ levels. Upon assessment of PHHs treated with CF, we observed a reduction in mRNA transcript levels of ER stress markers GRP78

**Fig. 3 Endoplasmic reticulum Ca$^{2+}$ modulates PCSK9 expression and secretion. A** HuH7 cells were either transfected with a FRET-based ER-resident Ca$^{2+}$ sensor, D1ER, or pre-loaded with the low-affinity Ca$^{2+}$ indicator, Mag-Fluo-4 (green color). Cells were subsequently treated with either thapsigargin (TG; 100 nM), CDN (100 μM) or caffeine (CF; 200 μM) for 24 h. Fluorescence intensity was measured using a fluorescent spectrophotometer and visualized using a fluorescent microscope ($n = 6$ biologically independent samples per group; data presented are mean ± s.d.). **B** HuH7 cells were pretreated with either CF or vehicle for 24 h and loaded with the high-affinity Ca$^{2+}$ dye, Fura-2-AM. Exposure of cells to a high dose of TG (1 mM) induced a spontaneous depletion of endoplasmic reticulum (ER) Ca$^{2+}$ (*, $p < 0.05$ vs. vehicle-treated). **C** The expression of an ER-resident Ca$^{2+}$ binding protein, calnexin, was examined in CF- and TG-treated HuH7 cells using immunoblots. **D–G** PCSK9, SREBP2, and GRP78 mRNA expression was assessed in HuH7 cells treated with a variety of ER Ca$^{2+}$ modulators including: ryanodine receptor agonist (ryanodine, 10 nM), ryanodine receptor antagonist (ryanodine, 10 μM), SERCA pump activator CDN (100 μM) and IP3R antagonist 2APB (50 μm), in the presence and absence of TG (100 nM) for 24 h. **H** mRNA transcript levels were also examined in HuH7 cells treated with SERCA pump inhibitors, TG (100 nM), and CPA (10 μM). **I–K** The effect of pharmacologic agents and plasmid-derived CMV-driven proteins, known to affect ER Ca$^{2+}$ levels, on secreted PCSK9 levels was then examined using ELISAs. **L**, **M** Secreted PCSK9 levels were also examined in TG- and U18-treated cells. **N** The effect of CF on secreted PCSK9 levels was also examined in cells incubated in Ca$^{2+}$-deficient medium. For panels **D–N**: $n = 5$ biologically independent samples per group; data presented are mean ± s.d. Scale bars; 200 μm. Statistical comparisons between two groups were conducted using unpaired two-tailed Student's $t$-tests, while multiple groups were compared using oneway ANOVAs with the Tukey HSD post hoc test (*$p < 0.05$; **$p < 0.01$; ***$p < 0.001$; ****$p < 0.0001$). Source data are provided as a Source Data file.

and the activating transcription factor 4 (ATF4; Fig. 4G). Similar experiments were also carried out in cultured HuH7 cells (Fig. S5A, B), in which a CF-mediated reduction in the expression of pPERK, IRE1α, sXBP1, ATF4 and ATF6 was also observed via immunoblotting and real-time PCR. Reactive oxygen species production, a process known to occur during conditions of ER stress, was also attenuated by CF (Fig. 5H). Consistent with its effect on ER stress markers, CF blocked the accumulation of thioflavin-T-stained misfolded protein aggregates and the expression of FLAG-sXBP1 in cells transfected with the ER activated indicator plasmid[34] (Fig. 4I–K). Collectively, these data support a model in which heightened ER Ca$^{2+}$ levels promote chaperone function and efficiency, thereby leading to a reduction in chaperone abundance. While increasing the protein binding ability of chaperones, such as GRP78, CF also attenuates SREBP2-driven gene expression (Fig. 4L).

**CF blocks hepatic ER chaperone expression and attenuates PCSK9 secretion in mice.** Next, we assessed the effect of CF on PCSK9 expression/secretion and UPR activation in mice. Following IP injection of CF (50 mg/kg − 8 h) we observed a significant reduction of circulating PCSK9 and triglyceride levels (Fig. 5A, B). A time-course experiment also revealed that CF treatment required 4 h to significantly reduce plasma PCSK9 levels in mice (Fig. 5C). Consistent with these observations, administration of CF via oral-gavage (20 mg/kg) also reduced circulating PCSK9 levels (Fig. S6) The protein and mRNA expression of UPR chaperones, GRP78, GRP94, IRE1α, and CHOP was also assessed via immunohistochemical staining, immunoblots and real-time PCR in the livers of these mice. Consistent with our findings in cultured cells, CF reduced the expression of UPR markers (Fig. 5D–G). Likewise, in a manner similar to previous reports, an inverse correlation between plasma PCSK9 levels and the expression of hepatic cell-surface LDLR and CD36 protein was observed (Fig. 5D–E)[2,4,35]. LDLR staining was also performed in $Pcsk9^{-/-}$ and $Ldlr^{-/-}$ mice to confirm the specificity of the antibodies used in our study (Fig. S7). Immunohistochemical staining intensities were quantified using ImageJ Software (Fig. 5E). Despite the increase in hepatic cell-surface LDLR levels, real-time PCR data revealed that LDLR transcript levels were reduced by CF; a result that is consistent with other SREBP-2 regulated genes. Similar to CF, CDN also increased hepatic LDLR expression in mice (Fig. S5C. 50 mg/kg; IP; 8 h).

To confirm that our mouse model was responding to treatments in a manner consistent with previous studies, mice were treated with alirocumab; a well-established clinically approved anti-PCSK9 monoclonal antibody[36]. Treatment with alirocumab led to a significant increase in hepatic LDLR expression (Fig. 5H, I). A reduction in the mRNA levels of SREBP2, PCSK9 and the LDLR was also observed (Fig. 5J). Overall, these studies demonstrate that CF blocks the secretion of PCSK9 and increases the expression of the LDLR in vivo.

**CF increases hepatic LDL uptake.** It is well-established that PCSK9 enhances the degradation of the LDLR and reduces the capacity of hepatocytes to bind and internalize extracellular LDLc[37]. We therefore postulated that CF and other agents that increase ER Ca$^{2+}$ levels, may also augment LDLc clearance. We started by confirming that CF increased the expression of PCSK9-regulated receptors in our cultured cell models using immunoblots (Fig. 6A). Next, we developed an assay whereby HepG2 cells plated in black clear-bottom 96-well plates were treated with agents for 24 h and subsequently exposed to fluorescently labeled DiI-LDL for 5 h in FBS-free medium prior to analysis. The uptake and accumulation of DiI-LDL was then quantified using a fluorescent spectrophotometer (Molecular Devices). Interestingly, we observed that CF increased LDLc uptake and that U18, an agent that increased secreted PCSK9 levels (Fig. 3M), reduced LDLc uptake (Fig. 6B). To confirm that CF increased LDLc uptake in cultured hepatocytes in a manner dependent on PCSK9 inhibition, this experiment was repeated in HepG2 cells stably transfected with PCSK9 shRNA (Fig. 6C). As expected, CF treatment failed to significantly increase LDLc uptake in conditions of reduced PCSK9 levels. Live-cell staining of the LDLR was also performed in HepG2 cells exposed to DiI-LDL (Fig. 6D). Increased cell-surface LDLR, as well as intracellular DiI-LDL, was observed in CF-treated HepG2 cells compared to vehicle-treated cells using a fluorescent microscope. Despite the increased protein abundance of the LDLR, as well as increased uptake of LDLc, CF reduced mRNA transcript levels of the LDLR in HuH7 and HepG2 cells (Fig. 6E). Given the observed increase in CD36 receptor levels, additional experiments were conducted to determine whether CD36 played a role in LDLc uptake in response to CF. Results from these experiments demonstrate that the knockdown of CD36 via siRNA (siCD36), as well as the pharmacologic inhibition using sulfosuccinimidyl oleate (SSO), failed to affect CF-mediated DiI-LDL uptake (Fig. 6F, G). The knockdown of CD36 was confirmed via immunoblotting (Fig. 6H).

We next examined the effect of CF on hepatic LDLc uptake in mice. Accordingly, $Pcsk9^{+/+}$ and $Pcsk9^{-/-}$ mice were treated with either CF or PBS-vehicle for 8 h, as well as fluorescently labeled DiI-LDL for 1 h prior to sacrifice. In support of our in vitro studies, we observed that CF increased hepatic cell-surface LDLR expression in the $Pcsk9^{+/+}$ mice but did not increase LDLR expression in $Pcsk9^{-/-}$ mice (Fig. 6I). CF also increased hepatic DiI fluorescence intensity and reduced serum

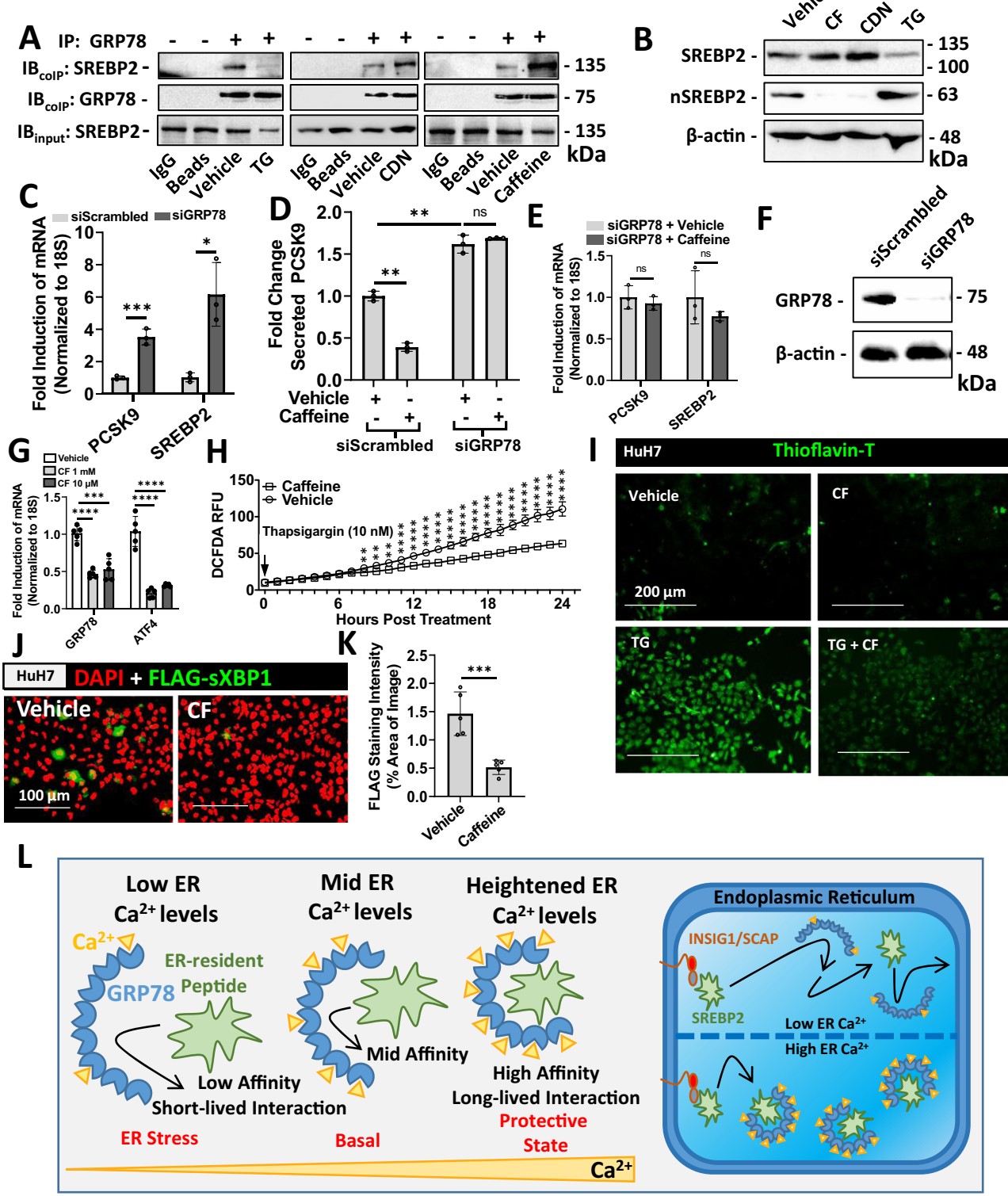

DiI fluorescence intensity in *Pcsk9*[+/+] mice but failed to affect these parameters in *Pcsk9*[−/−] mice (Fig. 6J). Hepatic cell-surface LDLR immunostaining and DiI-LDL uptake were also examined using a fluorescent microscope (Fig. 6K). Finally, native LDLc was also examined in 18-week-old male C57BL/6J mice treated with CF (30 mg/kg) every 24 h for 14 days (Fig. 6L, M). Using ELISAs, a reduction of the surrogate marker of LDLc, Apolipoprotein B (ApoB), as well as PCSK9 was observed in response to a daily dose of CF. Collectively, these data suggest that CF increases

hepatic LDLc clearance by increasing LDLR expression in a manner dependent on its ability to block PCSK9 secretion from hepatocytes.

**CF reduces plasma PCSK9 levels in healthy human subjects.** Given that CF is among the most commonly consumed pharmacologically active compounds in the world[8], we assessed its ability to affect PCSK9 levels in fasted healthy volunteers. Serum was collected prior to, as well as 2- and 4-h post CF treatment

**Fig. 4 Endoplasmic reticulum Ca$^{2+}$ regulates the interaction between GRP78 and SREBP2. A** HuH7 cells were treated with control agents thapsigargin (TG; 100 nM), which causes ER Ca$^{2+}$ depletion, or CDN (100 µM), a compound known to increase endoplasmic reticulum (ER) Ca$^{2+}$ levels. The effect of caffeine (CF; 200 µM) was also assessed. Following 24 h treatment, a co-immunoprecipitation (IP) for GRP78 was carried out. Protein loading was normalized to GRP78 and relative co-immunoprecipitated SREBP2 was examined via immunoblots (IB). **B** The effect of CF, CDN, and TG on the retention of ER-resident pre-mature SREBP2, and on the activated nuclear SREBP2 (nSREBP2), was also assessed via IB. (**C-E**) To confirm the role of GRP78 in CF-mediated PCSK9 inhibition, mRNA transcript and secreted protein levels were examined in HepG2 cells exposed to siRNA targeted against GRP78 (siGRP78) ($n = 3$ biologically independent samples per group; data presented are mean ± s.d.). **F** Knockdown of GRP78 was confirmed via IB. **G** ER stress markers were assessed in primary human hepatocytes (PHH) treated with CF (200 µM) and CDN (10 µM) via real-time PCR ($n = 5$ biologically independent samples per group; data presented are mean ± s.d.). **H** The effect of CF on reactive oxygen species production, resulting from the treatment of TG (100 nM), was also assessed in HuH7 cells ($n = 3$ biologically independent samples per group; data presented are mean ± s.d.). **I** ER stress-induced amyloid deposition was examined using the fluorescent stain, Thioflavin-T (green color). **J, K** HuH7 cells were transfected with the ER activated indicator plasmid encoding an ER stress-inducible FLAG-sXBP1 (green color; $n = 5$ biologically independent samples per group; data presented are mean ± s.d.). Staining intensity was quantified using ImageJ software. **L** Model in which Ca$^{2+}$ promotes the GRP78-mediated sequestration of SREBP2 in the ER. Statistical comparisons between two groups were conducted using unpaired two-tailed Student's $t$-tests, while multiple groups were compared using one-way ANOVAs with the Tukey HSD post hoc test (*$p < 0.05$; **$p < 0.01$; ***$p < 0.001$; ****$p < 0.0001$). Source data are provided as a Source Data file.

(400 mg orally; ~5 mg/kg). Consistent with our observations in cultured hepatocytes and in mice, CF reduced plasma PCSK9 levels in healthy subjects by 25% ($n = 12$) and 21% ($n = 8$) at the 2- and 4-h time points, respectively (Fig. 7A, B). Plasma PCSK9 levels were also examined in control subjects that did not consume CF, to verify whether the additional 2 h of fasting during the course of the experiment would alter PCSK9 levels. No significant difference was observed in this group ($n = 5$; Fig. 7C).

**Characterization of CF derivatives as antagonists of PCSK9.** Our data demonstrated that CF antagonized secreted PCSK9 levels in pre-clinical models, as well as in humans. Interestingly, previous studies have also characterized small molecules that block PCSK9[38,39]. CF, however, is a well-characterized compound having several health benefits with few known adverse side effects. Achieving an optimal level of PCSK9 inhibition absent of the neuro-excitatory effect of CF, however, may be a challenge for the long-term clinical application of these findings. To address this concern, a variety of caffeine derivatives were screened as potential alternatives to CF that may achieve significant PCSK9 inhibition while avoiding the undesired neuro-excitatory effect (Fig. 8A, B; see chemical structure of CF derivatives in Supplementary Table 2). Interestingly, CF metabolites including theobromine and paraxanthine, as well as other xanthine-derived compounds, such as PSB603, 8CD and 8CC, exhibited a dose-dependent reduction of mRNA expression and secreted levels of PCSK9 (Fig. 8A, B).

We have now initiated a medicinal chemistry program with the aim of developing new xanthine derivatives that are optimized for anti-PCSK9 activity. Here we disclose two new compounds, MLRA-1812 and MLRA-1820, that are a product of this effort (Fig. 8C–G). These compounds have significantly greater efficacy for PCSK9 inhibition than CF. Experiments done in HepG2 cells demonstrate that treatment with MLRA-1812 and MLRA-1820 yielded a two-fold reduction of secreted PCSK9 levels compared to CF treatment at the same dose (Fig. 8C). Moreover, at a concentration of 100 nM, these compounds achieved similar levels of inhibition as CF at a dose of 100 µM. Assessment at the mRNA level revealed a similar mode of action, whereby antagonism of SREBP2 reduced *de novo* synthesis of PCSK9 (Fig. 8D). Like CF, we also observed that MLRA-1812 and MLRA-1820 did not exhibit cytotoxic properties on cultured HepG2 cells (Fig. 8E). Next, the impact of MLRA-1812 and MLRA-1820 on DiI-LDL uptake was examined. Consistent with the observed reduction in secreted PCSK9 levels, treatment of HepG2 cells with these compounds led to a significant increase in LDLR expression (Fig. 8F) and DiI-LDL uptake (Fig. 8G). Overall, these data demonstrated that a variety of xanthines exhibit

potency for the antagonism of PCSK9 expression. The results of our ongoing comprehensive structure-activity relationship studies will be reported in due course.

## Discussion

The effects of CF on the vascular system and CVD have been examined by others in the past[8,9]. Given that CF consumption occurs primarily in the form of beverages that contain inconsistent doses and that are frequently mixed with adulterants such as dairy and sugar product, results from such studies can be difficult to interpret and often vary. A recent meta-analysis provides an in-depth summary of the current body of literature existing on the effects of CF consumption on cardiovascular outcomes, including total CVD[9]. Interestingly, the majority of studies examined, which involved thousands to hundreds of thousands of patients, demonstrated a protective effect of CF consumption against CVD risk.

CF is known to exert its effect through a range of molecular targets, including the antagonism of adenosine receptors, GABA receptors, and phosphodiesterase enzymes, as well as inducing intracellular Ca$^{2+}$ transients by enhancing RyR-mediated calcium-induced calcium release (CICR)[11]. Although the aforementioned interactions do not directly support our observation, in which CF increased ER Ca$^{2+}$ levels, CF is also known to block ER Ca$^{2+}$ release via inhibition of the IP3-receptor[40,41]. In addition, CF was previously shown to bind to hepatic RyR and potentially block RyR-mediated Ca$^{2+}$ release[42]. Given the broad range of targets known to interact with CF, the identification of exact molecular mechanisms pertaining to its protective effect on the vascular system is challenging. In support of the aforementioned protective effect of CF on CVD risk, CF has been shown to promote vasodilation of the vascular endothelium by means of stimulating Ca$^{2+}$ release via CICR and leading to the activation of eNOS[34]. In vascular smooth muscle cells, following CICR-induced vasodilation, CF also increases intracellular Ca$^{2+}$ levels by inducing non-selective cation channels at the cell surface[43] and has been shown to block IP3Rs[44]. These studies, as well as those of others[45,46], are consistent with our observations that CF increases intracellular Ca$^{2+}$ levels and attenuates ER stress. Collectively, however, there exists a range of mechanisms by which CF affects intracellular Ca$^{2+}$ levels, which tend to differ between tissue and/or cell types. In hepatocytes, our data strongly suggest that CF increases cytosolic and ER Ca$^{2+}$ levels following a 24 h exposure.

The ER serves as an important and dynamic Ca$^{2+}$ reserve, capable of extruding Ca$^{2+}$ for signaling and/or excitatory purposes and removing excess cytosolic Ca$^{2+}$ following periods of excitement. The Ca$^{2+}$ sequestering capacity of the ER, which far

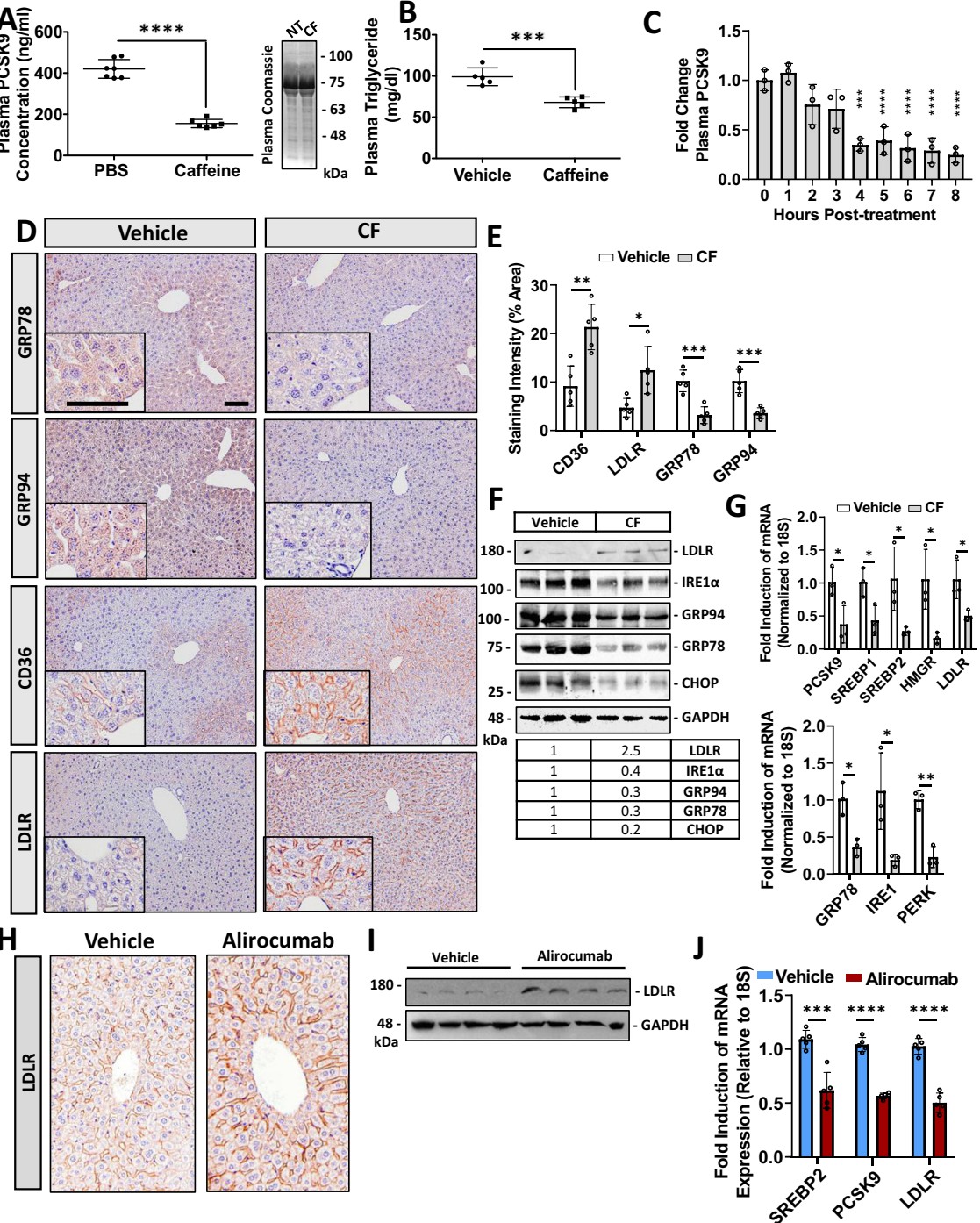

**Fig. 5 Caffeine reduces chaperone expression and blocks hepatic PCSK9 expression in mice.** 12-week-old male C57BL/6J mice were treated with caffeine (CF; 50 mg/kg) and fasted for 8 h prior to sacrifice (*n* = 6). **A**, **B** Plasma PCSK9 and triglyceride levels were measured using an ELISA and colorimetric assays, respectively (*n* = 6 biologically independent samples per group; data presented are mean ± s.d.). **C** The time-dependence of CF on plasma PCSK9 levels was also determined using an ELISA (*n* = 5 biologically independent samples per group; data presented are mean ± s.d.). **D** The livers of these mice were assessed for cell-surface expression of LDLR and CD36, as well as the ER stress markers GRP78 and GRP94 via immunohistochemical staining (*n* = 5). **E** Staining was quantified using ImageJ software (*n* = 5 biologically independent samples per group; data presented are mean ± s.d.). **F**, **G** The expression of ER stress markers (GRP78, PERK, and IRE1α) as well as cholesterol-regulatory genes (LDLR, PCSK9, HMGR, SREBP1 and SREBP2) were also examined using immunoblots and real-time PCR (*n* = 5 biologically independent samples per group; data presented are mean ± s.d.). **H–I** 12-week-old male C57BL/6J mice were treated with a single subcutaneous injection of the anti-PCSK9 neutralizing antibody, alirocumab (30 mg/kg), for 10 days (*n* = 10). LDLR expression was assessed using immunohistochemistry and immunoblots. **J** The mRNA expression of SREBP2, PCSK9, and the LDLR was assessed via real-time PCR (*n* = 5 biologically independent samples per group; data presented are mean ± s.d.). Bars; 50 μm. Statistical comparisons between two groups were conducted using unpaired two-tailed Student's *t*-tests, while multiple groups were compared using one-way ANOVAs with the Tukey HSD post hoc test (**p* < 0.05; ***p* < 0.01; ****p* < 0.001; *****p* < 0.0001). Source data are provided as a Source Data file.

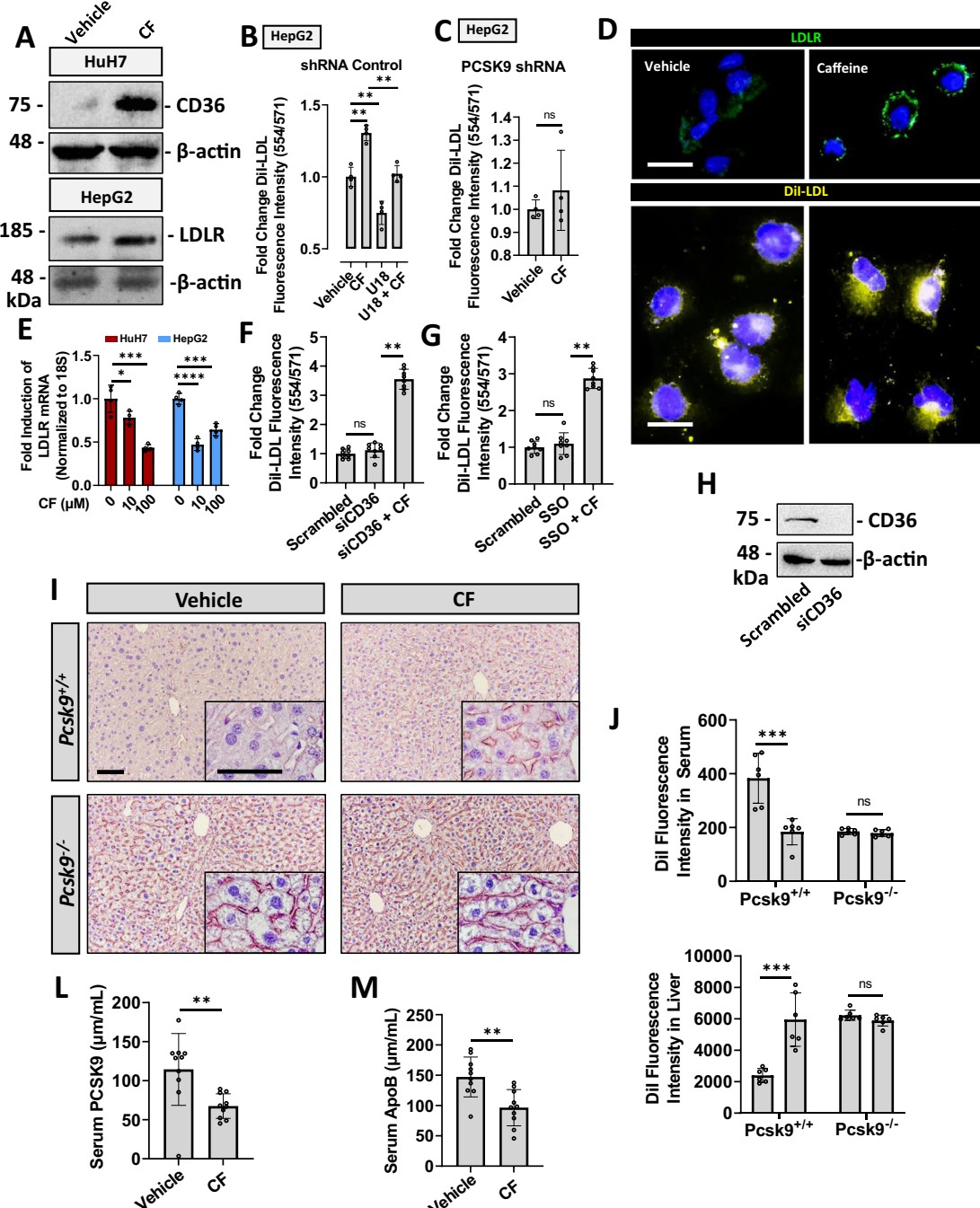

**Fig. 6 Caffeine increases hepatic LDL uptake in a PCSK9-dependent manner. A** The expression of PCSK9-regulated receptors, LDLR and CD36, was examined in caffeine (CF)-treated cultured hepatocytes (200 μM). **B** The uptake and intracellular accumulation of fluorescently labeled DiI-LDL was examined in cells treated with CF in the presence or absence of the PCSK9-inducer, U18, using a fluorescent spectrophotometer ($n = 4$ biologically independent samples per group; data presented are mean ± s.d.). **C** The effect of CF treatment (200 μM) on DiI-labeled LDL uptake was also examined in PCSK9 shRNA knockdown cells ($n = 4$ biologically independent samples per group; data presented are mean ± s.d.). **D** Immunofluorescent staining of cell-surface LDLR was carried out in live CF pre-treated HepG2 cells (200 μM). Cellular DiI-LDL accumulation was also visualized in CF-treated HepG2 cells using a fluorescent microscope. **E** Expression of the LDLR in CF-treated HuH7 and HepG2 cells was measured via real-time PCR ($n = 4$ biologically independent samples per group; data presented are mean ± s.d.). **F–G** The uptake of DiI-LDL was quantified in HepG2 cells transfected with siRNA targeted against CD36 and a pharmacologic inhibitor of CD36 (SSO) (10 μM) ($n = 8$ biologically independent samples per group; data presented are mean ± s.d.). **H** Knockdown was confirmed via immunoblotting. $Pcsk9^{+/+}$ and $Pcsk9^{-/-}$ mice were treated with either PBS-vehicle or CF, as well as fluorescently labeled DiI-LDL (1 μg/kg). **I–K** Hepatic cell-surface LDLR expression was assessed via immunohistochemistry (DAPI: blue; LDLR: green; DiI-LDL: red). **J** Hepatic and serum DiI-LDL fluorescence intensity was quantified using a fluorescent spectrophotometer and visualized using a fluorescent microscope ($n = 6$ biologically independent samples per group; data presented are mean ± s.d.). **L, M** Native LDLc was also examined in 18-week-old male C57BL/6J mice treated with CF (30 mg/kg) every 24 h for 14 days via ELISA of the surrogate marker ApoB; serum PCSK9 levels were also assessed via ELISA ($n = 10$ biologically independent samples per group; data presented are mean ± s.d.). Scale bars: **D** 10 μm; **I** 50 μm; **K** 100 μm. Statistical comparisons between two groups were conducted using unpaired two-tailed Student's $t$-tests, while multiple groups were compared using one-way ANOVAs with the Tukey HSD post hoc test (*$p < 0.05$; **$p < 0.01$; ***$p < 0.001$; ****$p < 0.0001$). Source data are provided as a Source Data file.

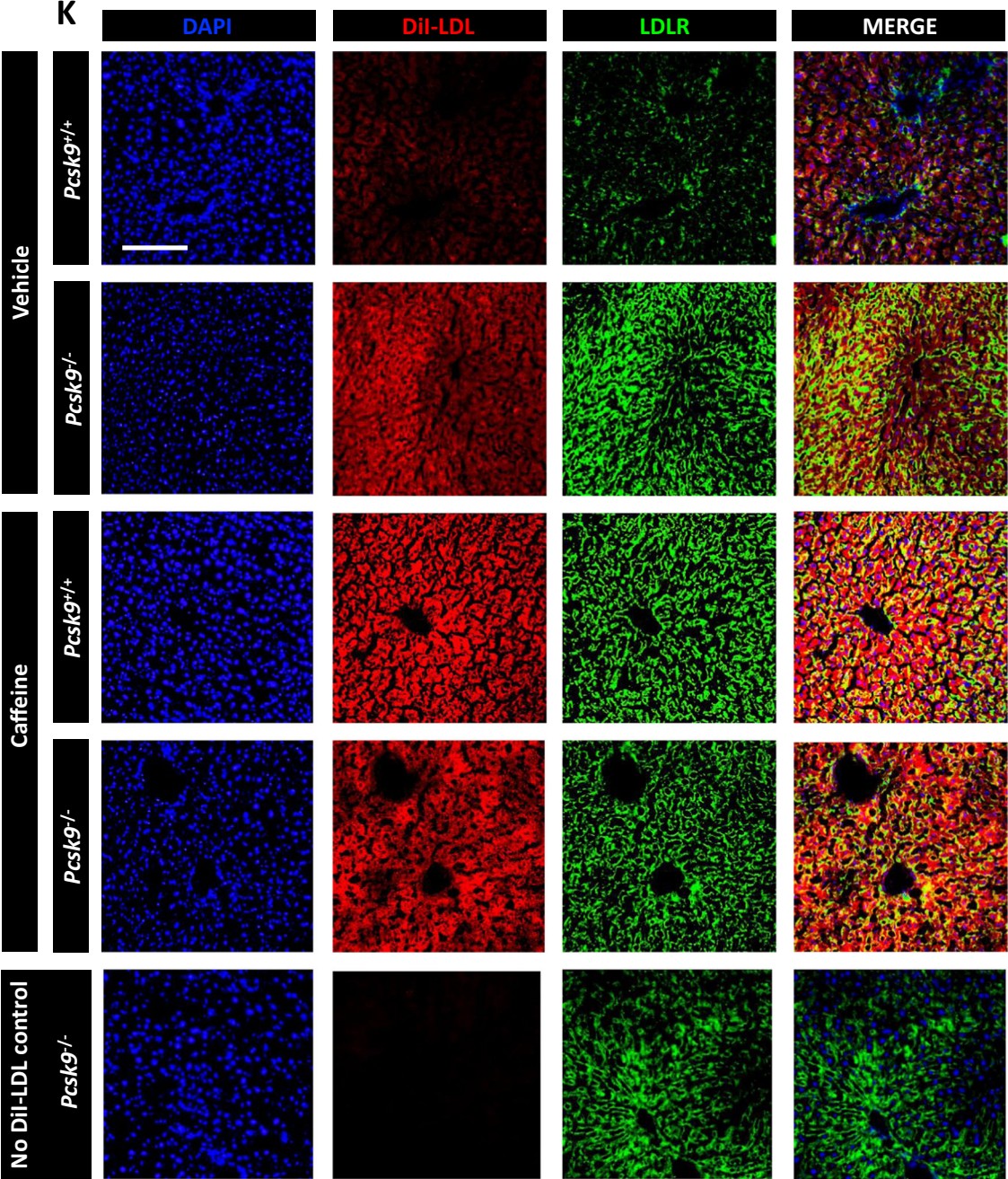

**Fig. 6** (Continue)

exceeds that of the cytosol, depends on the abundance of ER-resident low-affinity/high capacity $Ca^{2+}$-binding proteins[47]. The interaction of $Ca^{2+}$ with these proteins also promotes chaperone function. During conditions of ER $Ca^{2+}$ depletion, chaperones lose their folding capacity and misfolded polypeptides accumulate in the ER. The UPR, which increases the abundance of ER-resident chaperones, is then triggered in order to restore ER folding capacity and ER $Ca^{2+}$ levels[48]. Conversely, increasing ER $Ca^{2+}$ levels appears to have a net protective effect on ER homeostasis, given the observed reduction in UPR marker expression by agents that activate SERCA and increase ER $Ca^{2+}$ influx or those capable of blocking leakage from either IP3R or RyR[21,49]. Consistent with this observation, we also found that CF protected against TG-induced ER stress in cultured hepatocytes and reduced the expression of a variety of ER chaperones in the livers of mice.

In addition to UPR chaperones, ER stress is also known to promote the activation of the self-induced transcription factors that regulate the expression of fatty acid- and cholesterol-regulatory genes, namely *SREBP1* and *SREBP2*[50]. Although the exact mechanism by which ER stress activates the SREBPs remains elusive[15], previous studies have suggested and/or demonstrated that (a) ER stress can reduce the expression of *INSIG1*, a negative regulator of the SREBPs[51], (b) ER stress-induced caspases can cleave and activate the SREBPs in a manner independent of conventional S1P activation in the Golgi and[52,53], and (c) GRP78 can dissociate from the SCAP-SREBP complex thus liberating SREBP from the constraints of the ER[54]. We now report the finding that ER $Ca^{2+}$ levels serve to fine-tune the peptide-binding capacity of GRP78, thereby affecting the ER retention of its binding partners, such as pre-mature SREBP2.

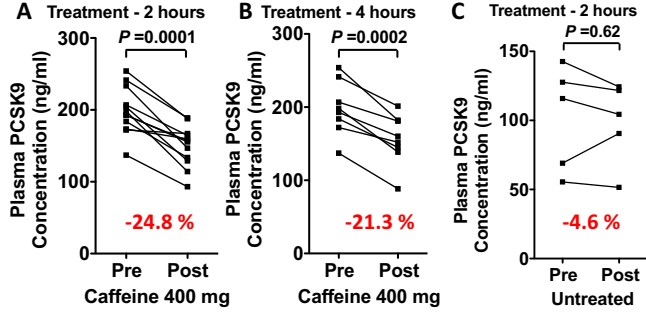

**Fig. 7 Caffeine reduces plasma PCSK9 levels in healthy volunteers.**
**A, B** Healthy subjects between the ages of 22 and 45 years were administered 400 mg (~ 5 mg/kg) of caffeine (CF) following a 12 h fasting period. Plasma PCSK9 levels were measured before administration, as well as 2- and 4 -h following administration ($n = 12$ and $n = 5$, respectively). **C** PCSK9 levels were also measured in a group of individuals ($n = 5$) that were not administered CF to control for the additional 2 h of fasting time during the experiment. Differences between groups were determined using a paired two-tailed Student's $t$-test.

We also observed that CF induced the protein expression of the LDLR and increased LDLc uptake in cultured hepatocytes. Given that SREBP2 regulates *de novo* expression of *PCSK9* and the *LDLR*, the observed induction of cell-surface LDLR in the face of SREBP2 inhibition likely occurs in response to the loss of circulating PCSK9 levels. Several studies have demonstrated that, with a half-life of five minutes[55], PCSK9 expression closely follows that of SREBP2[56]. In contrast, with a 144-fold increased half-life compared to PCSK9 (12 h)[57], expression of the LDLR appears less dependent on *de novo* synthesis and more on factors that influence its stability at the cell surface, like circulating PCSK9. Because SREBP2 also induces the expression of HMGR, it is also possible that CF reduces circulating LDLc levels via inhibition of HMGR-mediated *de novo* cholesterol synthesis. It is well-established, however, that statins block HMGR, induce SREBP2 activity and increase circulating PCSK9 levels. In contrast to statins, our findings demonstrate that CF blocks both SREBP2 and PCSK9. Thus, with reduced circulating PCSK9 levels, it is unlikely that HMGR activity is increased in response to CF[58].

PCSK9 enhances degradation of the LDLR and promotes the onset and progression of CVD, which represents one of most challenging and costly health care problems that society faces today[59]. Developing our understanding of the regulatory mechanisms that modulate the expression and secretion of PCSK9 from hepatocytes may aid in the development of anti-PCSK9 therapies that are less costly than those that are currently available. Overall, results from our study support a model in which small molecules like CF, capable of increasing ER $Ca^{2+}$ levels, can block the activation of SREBP2 by enhancing GRP78 chaperone function and binding capacity (Fig. 9). We also report that CF potently blocks the expression of PCSK9, a downstream target of SREBP2 transcriptional activity, in cultured hepatocytes, in mice, and in healthy human subjects. By extension, we also observed that CF induced the expression of cell-surface hepatic LDLR and increased the uptake of LDLc. Our findings delineate a mechanism by which ER $Ca^{2+}$ and its modulators can affect the expression and activity of proteins that play a central role in CVD. Overall, this study provides compelling evidence that the xanthine scaffold is a potent starting point for the development of compounds capable of mitigating CVD risk.

## Methods
**Cell culture, treatments, and transfections**. HuH7 (kind gift from Dr. Nabil G. Seidah) and HepG2 (ATCC; HB-8065) cells were routinely grown in complete

Dulbecco's Modified Eagle's Medium (Gibco, Thermofisher Scientific) supplemented with 10% fetal bovine serum (Sigma-Aldrich) and 100 U/ml of penicillin and streptomycin (Sigma-Aldrich). CF, ryanodine, 2 APB, CDN, theobromine, paraxanthine, 8-cyclopentyl-1,3-dimethylxanthine (8CD), 8-(3-Chlorostyryl) CF (8CC), PSB603, cyclopiazonic acid and U18666A were purchased from Tocris Bioscience. All cell treatments were carried out for 24 h unless otherwise stated. Cells were transfected with a cocktail consisting of plasmid DNA (1 µg), X-tremeGENE HP (3 µl; Thermo Fisher Scientific), and opti-MEM (100 µl; Thermo Fisher Scientific) per 1 ml complete medium containing plated cells. Human PCSK9 was overexpressed using the bicistronic pIRES-EGFP plasmid; calnexin using the mPA-GFP-N1 plasmid. To attenuate the expression of GRP78 and CD36, siGENOME smartpool siRNA was purchased from GE Dharmacon (M-008198-02 and L-010206-00-0005 respectively) and transfected using lipofectamine RNAi-MAX as per the manufacturer's protocol.

**Ca$^{2+}$ studies: fluorogenic dyes and genetically encoded FRET-based sensors**. Intracellular $Ca^{2+}$ in Huh7 and HepG2 cells was measured using a high-affinity $Ca^{2+}$ indicator, Fura-2-AM (Thermo Fisher Scientific). ER $Ca^{2+}$ levels were assessed using the low-affinity $Ca^{2+}$ indicator, Mag-Fluo-4, and via transfection of cells with D1ER. The D1ER plasmid encodes an ER-resident calcium binding protein linked to a fluorescent protein and increases in fluorescence intensity upon $Ca^{2+}$ binding[7]. For assessment using indicators, cells were plated in black clear-bottom 96-well plates to a confluence of 70–75% and treated with $Ca^{2+}$ modulating agents for 24 h ($n = 6$). Cells were then washed and incubated with Fura-2-AM (2 µM) or Mag-Fluo-4 (2 µM) for 45 min at 37 ˚C in HBSS containing 20 mM HEPES and 2% pluronic acid v/v (Thermo Fisher Scientific). Fluorescence intensity of intracellular Fura-2-AM was measured at two distinct wavelengths (ex 340/em 515 and ex 380/em 515), following three consecutive washes, to assess bound and unbound states using a SpectraMax GeminiEM fluorescent spectrophotometer (Molecular Devices, Sunnyvale, California, USA). Fluorescence intensity of Mag-Fluo-4 was quantified at a single wavelength (ex 495/em 515). For assessment using D1ER, HuH7 cells were plated in black clear-bottom 96-well plates to a confluence of 70–75% and transfected ($n = 6$). Twenty-four hours later, cells were treated with $Ca^{2+}$ modulating agents for an additional 24 h and quantified (ex 495/em 515) using a SpectraMax GeminiEM fluorescent spectrophotometer.

**Immunoblot analysis**. Cells were washed in phosphate-buffered saline (PBS), lysed in 4X SDS-PAGE lysis buffer and separated on 7–10% polyacrylamide gels in denaturing conditions. Gels were transferred to nitrocellulose membranes using the BioRad mini trans-blot system, blocked in 5% skim milk in tris-buffered saline (TBS) for 1 h and incubated in primary antibody overnight for 16 h at 4 ˚C. Membranes were then exposed to horse radish peroxidase conjugated secondary antibodies and visualized using EZ-ECL chemiluminescent reagent (Froggabio). Band intensities were quantified using ImageJ software (BioRad) against membranes reprobed for housekeeping proteins. A complete list of antibodies used for immunoblot analysis is presented in Supplementary Table 3 of the supplementary materials.

**Immunoprecipitations**. Cells were grown in 10 cm dishes were resuspended in ice cold non-denaturing immunoprecipitation buffer containing 20 mM Tris HCL, 137 mM NaCl, 1% NP-40, 2 mM EDTA, and protease inhibitors (Roche). Total cell protein was normalized using a protein assay (BioRad) and 1 mg of protein from each sample was incubated with 2 µg of capture antibody targeted against GRP78 (Santa Cruz Biotechnology; SC-1050) and rotated on a platform for 24 h at 4 ˚C. Following incubation, samples were exposed to 100 µl of Protein G magnetic Surebeads (BioRad) for an additional 2 h on a rotating platform at 4 ˚C. Beads conjugated to the anti-GRP78 antibody were subsequently isolated and the remaining sample was collected and labeled "input" for use as controls. The magnetic beads underwent four consecutive washes using the non-denaturing IP buffer and resuspended and boiled in 100 µl of 4X SDS-PAGE sample buffer.

**Immunofluorescent Staining**. Cells were plated in 4-well chamber slides and incubated in complete DMEM for 24 h and exposed to treatments 24 h later for an additional 24 h. Cells were fixed with 4% paraformaldehyde for 30 minutes and washed with either non-permeabilizing, or permeabilizing PBS containing 0.025% Triton-X. Cells were then blocked with 1% bovine serum albumin (BSA) for 30 min and stained with anti-GFP antibody for 1 h in PBS containing 1% BSA. Afterwards, cells were washed and incubated with Alexa 488 fluorescently labeled secondary antibodies (Thermo Fisher Scientific), as well as the DAPI nuclear stain. Slides were then mounted with permafluor and visualized using the EVOS FL color imaging system at either ×20 or ×40 magnification. A complete list of antibodies used for immunofluorescence staining is presented in Supplementary Table 3 of the supplementary materials.

**Thioflavin-T staining**. Following treatment, live cells were incubated in complete DMEM containing 5 µM Thioflavin-T (ThT; Thermo Fisher Scientific) for 15 min. Cells were then fixed in 4% paraformaldehyde and mounted with permafluor. Fluorescent staining was visualized using the EVOS FL color system at either ×20 or ×40 magnification.

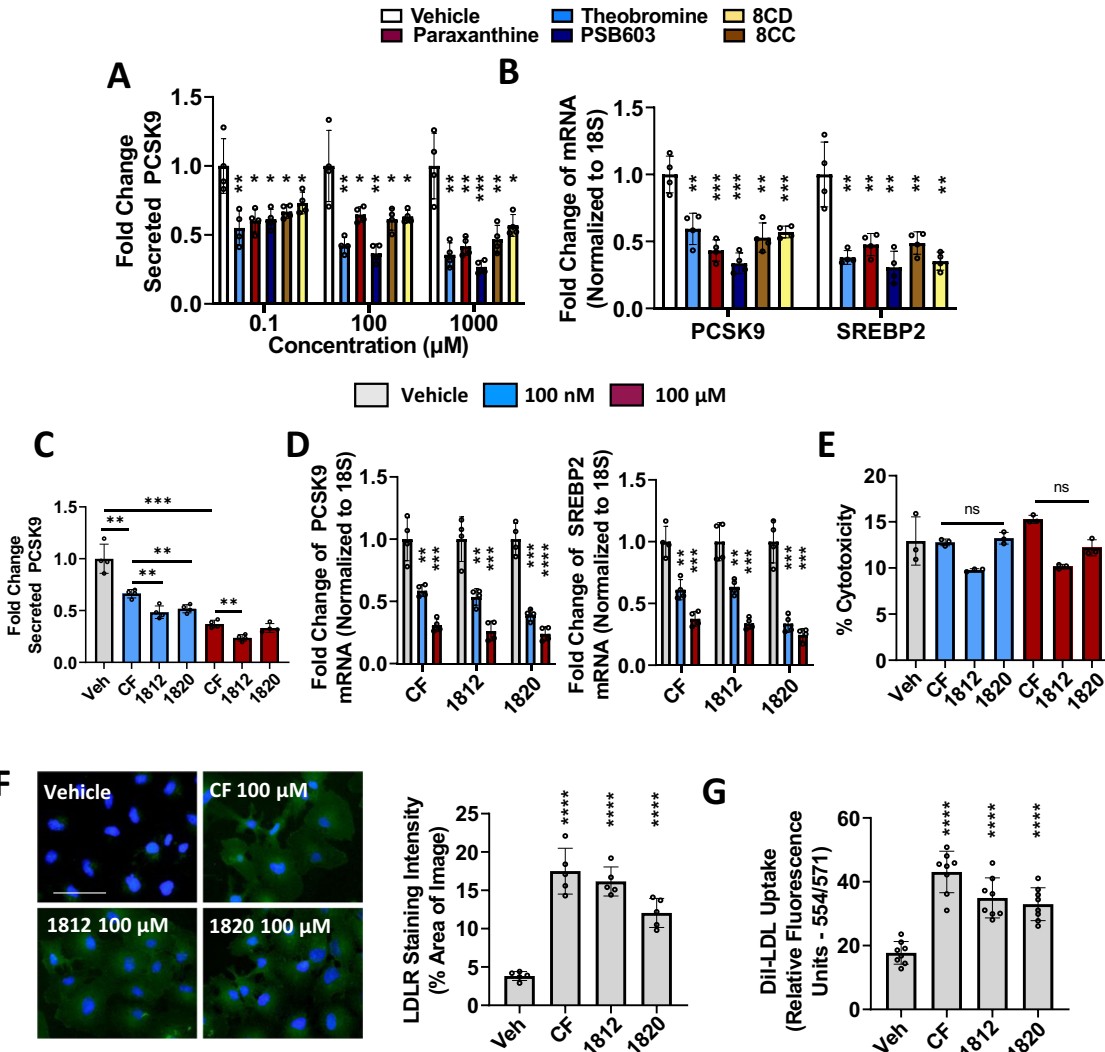

**Fig. 8 Characterization of xanthine-derived compounds as PCSK9 inhibitors. A**, **B** HepG2 cells were treated with increasing doses of caffeine (CF) metabolites, paraxanthine, and theobromine, as well as xanthine derivatives PSB603, 8CD, and 8CC. Secreted PCSK9 levels were assessed using ELISAs and mRNA transcript via real-time PCR. **C**, **D** Cells were treated with CF, as well as MLRA-1812 and MLRA-1820. Secreted PCSK9, as well as PCSK9 mRNA and SREBP2 mRNA were assessed in these cells. **E** The cytotoxicity of these agents was examined using an LDH assay. **F** HepG2 cells were treated with CF, MLRA-1812 (100 μM), and MLRA-1820 (100 μM) and assessed for cell-surface LDLR expression via immunofluorescent staining (green color); staining intensities were quantified using ImageJ software. **G** The uptake of DiI-LDL was also quantified in these cells using a spectrophotometer. *$p < 0.05$ vs. vehicle; Statistical comparisons between two groups were conducted using unpaired two-tailed Student's t-tests, while multiple groups were compared using one-way ANOVAs with the Tukey HSD post-hoc test (*$p < 0.05$; **$p < 0.01$; ***$p < 0.001$; ****$p < 0.0001$). Source data are provided as a Source Data file.

**Immunohistochemical staining**. Liver tissues were fixed in formaldehyde and subsequently embedded in paraffin for sectioning. Four μM thick sections underwent epitope retrieval and were subsequently stained with primary antibodies for 16 h at 4 °C. Slides were then exposed to biotin-conjugated secondary antibodies for 45 min and then streptavidin peroxidase for 10 min. Staining was visualized using the Nova Red HRP Substrate (Vector Laboratories). A complete list of antibodies used for immunohistochemical analysis is presented in Supplementary Table 3 of the Supplementary materials.

**Quantitative real-time PCR**. RNA purification/isolation was performed using RNeasy mini kits (Qiagen) and normalized to 2 μg RNA using a NanoDrop spectrophotometer. Samples were then reverse transcribed into cDNA using Superscript Vilo cDNA Synthesis kit (Thermo Fisher Scientific). Real-time PCR was executed with Fast SYBR Green (Thermo Fisher Scientific) using the ΔΔct method on the ViiA7 real-time PCR platform (Thermo Fisher Scientific). A complete list of primers used for PCR analysis is presented in Supplementary Table 4 of the supplementary materials.

**ELISAs**. Secreted PCSK9 levels were assessed directly in cell culture medium of cells grown in FBS-free medium for 24 h or in the serum isolated from either mice or human subjects. Mouse PCSK9 was measured using the Quantikinine ELISA kit (#MCP900, R&D Systems) and human PCSK9 using the PCSK9 Quantikinine ELISA kit (#DCP900, R&D Systems). Serum ApoB levels were also quantified using ELISAs (#DAPB00, R&D Systems). Serum samples were diluted as per manufacturer's instructions.

**Mouse studies and primary hepatocyte isolation**. All animal studies were carried out in 12-week-old, 8 h-fasted, male wild-type, *Pcsk9*−/− or *Ampkβ1*−/− mice on the C57BL/6J background and were performed in strict accordance with the McMaster University animal care guidelines and approved by the McMaster University ethics board. Mice were housed in 12:12 hour light:dark cycles and fed regular chow *ad libitum* (2918, Envigo). CF (25–100 mg/kg − 8 h) and CDN (50 mg/kg) treatments were administered via intraperitoneal injection unless specified otherwise. Primary mouse hepatocytes were isolated using a two-step process with EGTA (500 mM in HEPES buffer, Sigma Aldrich) and collagenase (0.05% in HEPES buffer, Sigma Aldrich) in 12-week-old male mice on the C57BL/6J

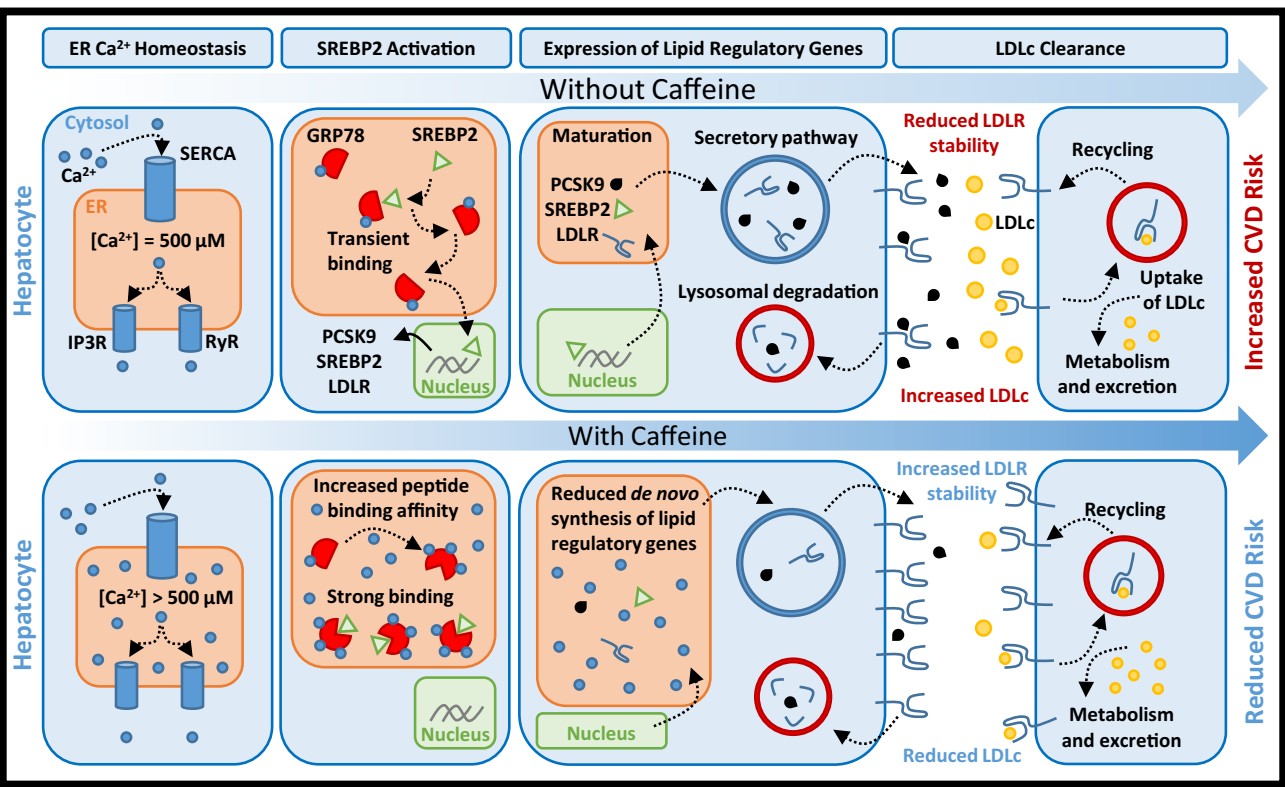

**Fig. 9 Caffeine blocks PCSK9 expression and increases LDLc clearance in hepatocytes.** The treatment of liver hepatocytes with caffeine increases the concentration of ER $Ca^{2+}$. Excess ER $Ca^{2+}$ leads to an increase in the peptide binding capacity and chaperone activity of ER-resident GRP78. The result is an ER-resident GRP78-SREBP2 complex with enhanced stability. The failure of SREBP2 to quickly exit the ER leads to a net reduction in expression of lipid regulatory genes, including PCSK9, SREBP2 and PCSK9. With reduced outflow of *de novo* PCSK9, cell-surface LDLR exhibits increased half-life and abundance, leading in a net increase in LDLc clearance.

background. Cells were then washed, centrifuged, and plated following isolation in cell strainers. Hepatosure 100-donor pooled primary human hepatocytes were purchased from Xenotech. Primary hepatocytes were regularly grown in William's E medium (Gibco, Thermo Fisher Scientific) containing 10% fetal bovine serum, 100 IU/ml penicillin, and 100 μg/ml streptomycin.

**DiI-LDL uptake assay.** Cells were seeded plated in black clear-bottom 96-well plates for 24 h and treated with experimental agents for an additional 24 h. During the last 5 h of treatment (19–24 h) cells were exposed to DiI-LDL (10 μg/ml) and then washed with two changes of pre-warmed (37 ˚C) HBSS containing 20 mM HEPES prior to analysis. The intracellular fluorescence intensity of DiI was then quantified using the SpectraMax GeminiEM fluorescent spectrophotometer (Molecular Devices; ex 554/em 571).

**CF studies in healthy human subjects.** Healthy human subjects between the ages of 22 and 45 ($n = 8$ males; $n = 4$ females) underwent fasting for 12 h prior to oral administration of 400 mg CF (~5 mg/kg, Wake ups). Blood was collected prior to administration and at 2 and 4 h following administration. All subjects completed and signed consent forms and studies were approved by the Hamilton Integrated Research Ethics Board, project number 5805. The subjects were not compensated for their participation in the study.

**Synthesis of MLRA-1812.** To a solution of 5,6-diamino-1,3-dimethyluracil (213.0 mg, 1.25 mmol, 1 equiv) in to a solution of MeCN/H2O (9:1, 5 mL) were added 3,5-bis(trifluoromethyl)benzaldehyde (1.25 mmol, 1 equiv), azobisisobutyronitrile (AIBN, 4.1 mg, 0.025 mmol, 0.02 equiv), and n-bromosuccinimide (NBS, 155.7 mg, 0.875 mmol, 0.7 equiv). The reaction mixture was stirred at 30 °C for 3 h. A precipitate was filtered and rinsed ethyl acetate ($2 \times 5$ mL) and methanol ($3 \times 5$ mL) to obtain the crude product. Chromatography over silica gel using gradient elution with DCM and methanol offered MLWK-1812 (8-(3,5-bis(trifluoromethyl)phenyl)-1,3-dimethyl-3,7-dihydro-1H-purine-2,6-dione, 196.1 mg, 0.50 mmol, 40% yield) as a white solid. Mp: 329-330o C. 1H NMR (700 MHz, DMSO-d6): d (ppm) 14.38 (s, br, 1H, NH), 8.76 (s, 1H), 8.24 (s, 1H), 3.53 (s, 3H, CH3), 3.28 (s, 3H, CH3). 13 C NMR DEPTQ (176 MHz, CDCl3): d (ppm) 154.32, 151.09, 148.14, 146.17, 131.38-130.81, 126.39, 125.41-120.76, q (J = 273.1 Hz),

123.31, 29.86, 27.84. FT-IR (cm-1): 3052.09, 2943.32, 2759.41, 1704.53, 1649.52, 1601.04. LCMS (ESI) m/z: 393.07807 calculated for ([M + H] + ); 393.07684 observed. The general procedure for the synthesis of MLRA-1812 was based on Bandyopadhyay et al.[60].

**Synthesis of MLRA-1820.** A reaction procedure analogous to that described for MLRA-1812 was followed using 2,4,5-trifluorobenzaldehyde to yield MLRA-1820 (1,3-dimethyl-8-(2,4,5-trifluorophenyl)-3,7-dihydro-1H-purine-2,6-dione, 117.9 mg, 0.38 mmol, 30% yield) as a light brown solid. Mp: 361–363 ºC. 1H NMR (700 MHz, DMSO-d6): δ (ppm) 13.86 (s, br, 1H, NH), 8.05–8.01 (dd, J = 17.1, 9.0 Hz, 1H), 7.83-7.79 (td, J = 10.4, 6.8 Hz, 1H), 3.49 (s, 3H, CH3), 3.27 (s, 3H, CH3). FT-IR (cm-1): 3281.65, 3039.61, 1695.87, 1651.37, 1602.14. LCMS (ESI) m/z: 311.07504 calculated for ([M + H]+); 311.07481 observed. The general procedure for the synthesis of MLRA-1812 was based on Bandyopadhyay et al.[60].

**Statistics.** Statistical comparisons between two groups were conducted using unpaired Student's t-tests, while comparisons between multiple groups were compared using one-way ANOVAs with the Tukey HSD post-hoc test. The paired Student's t-test was used to compared pre- and post-treatment values in human subjects. Differences between groups were considered significant at $p < 0.05$ (*$p < 0.05$; **$p < 0.01$; ***$p < 0.001$; ****$p < 0.0001$) and all values are expressed as mean ± s.d. Data from studies conducted in cell lines and mice are representative of at least three independent experiments.

**Reporting summary.** Further information on research design is available in the Nature Research Reporting Summary linked to this article.

## Data availability

All data generated in this study are available within the Article, Supplementary Information and Source Data files. Source data are provided with this paper.

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

## Acknowledgements

pcDNA-D1ER was a gift from Amy Palmer & Roger Tsien (Addgene plasmid # 36325) and mPA-GFP-Calnexin-N-14 was a gift from Michael Davidson (Addgene plasmid # 57122). This work was supported in part by research grants to R.C.A. from the Heart and Stroke Foundation of Canada (G-13-0003064 and G-15-0009389), the Canadian Institutes of Health Research (FRN173520), to N.G.S. from the Leducq Foundation (13 CVD 03), CIHR Foundation grant (148363) and Canada Research Chair (216684). G.R.S. is supported by a CIHR Foundation Grant (201709FDN-CEBA-116200) and a Tier 1 Canada Research Chair and J. Bruce Duncan Endowed Chair in Metabolic Diseases. Financial support from the Research Institute of St. Joseph's Healthcare Hamilton and Amgen Canada is acknowledged. R.C.A. is a Career Investigator of the Heart and Stroke Foundation of Ontario and holds the Amgen Canada Research Chair in the Division of Nephrology at St. Joseph's Healthcare and McMaster University.

## Author contributions

P.F.L., J.H.B., K.P., and R.C.A. conceived the studies and designed the experiments. In vitro studies were conducted by P.F.L. and J.H.B. In vivo studies were conducted by P.F.L. K.P. and M.E.M. Clinical studies were orchestrated by P.F.L. and J.H.B. C.F. derivatives were designed and developed by P.S., M.S., and J.M. The manuscript was written by P.F.L. and R.C.A. and was revised by J.H.B., K.P., G.P., G.R.S., L.J.J., S.R.W.C., N.G.S., S.A.I., M.A.T., and J.M.

## Competing interests

The authors declare the following competing interests: P.F.L., J.H.B, P.S., M.S., J.M., and R.C.A. have filed an institution-owned patent, entitled: "Compounds for Reducing Cholesterol and Treating Liver and Kidney Disease" (18-069_USProv) that relates to the development of caffeine and methylxanthine derivatives to lower cholesterol. R.C.A., J.M., G.P., and M.A.T. have an equity stake in Systemic Therapeutics. G.P. has received honoraria from Amgen and Sanofi. G.R.S. is a co-founder and shareholder of Espervita Therapeutics, a company developing new medications for liver cancer. McMaster University has received funding from Espervita Therapeutics, Esperion Therapeutics, Poxel Pharmaceuticals and Novo Nordisk for research conducted in the laboratory of G.R.S. G.R.S. has received consulting/speaking fees from Astra Zeneca, Eli Lilly, Esperion Therapeutics, Merck, Poxel Pharmaceuticals and Takeda. M.A.T. is the CEO of Exerkine Corporation and Stayabove Nutrition and is actively evaluating and marketing multi-nutrient supplements for the treatment of aging, obesity, muscular dystrophy and mitochondrial disease. No other authors declare any competing interests.

## Additional information

**Peer review information** *Nature Communications* thanks Huang-Ge Zhang, Julien Diana and the other anonymous reviewer(s) for their contribution to the peer review this work.

