## [Peer Review File · Nature Communications]

Reviewers' Comments:

Reviewer #1:

Remarks to the Author:

This study investigates the effect of caffeine in regulating cholesterol metabolism in vitro and in vivo. The authors show that CF suppress SREBP2 activation, reducing SREBP2-regulation of PCSK9 expression. While the finding is of interest, there are key experiments missing to support authors conclusion. In particular the authors should investigate the effect of CF in regulating the post-transcriptional regulation of LDLR (see below).

1- One of the major problems of the study is the lack of the experiments assessing the effect of caffeine (CF) in the posttranscriptional regulation of LDLR. Indeed, the authors did not assess the effect of CF in regulating LDLr mRNA levels. Both PCSK9 and LDLR are regulated by SREBP2, thus if CF regulates PCSK9 via SREBP2 activity and processing, LDLR mRNA should be elevated as well. However, the authors did not analyze the effect of CF on LDLR mRNA expression. Thus, additional experiments showing the post-transcriptional regulation of LDLr in vitro (e.g., actinomycin treatment etc) using CF are critical to support the mechanism proposed by the authors.

2- Is CF affecting PCSK9 secretion independent of the transcription?

3- The staining for hepatic LDLr looks non-specific (only stain the liver sinusoidal ECs). Please add a control for this staining using the LDLR null mice (Figure 5 and 6)

4- WB for LDLr in Figure 5F should be repeated (poor quality and overexposed)

5- CD36 also bind and uptake native LDL. Thus, it is not clear whether the increase uptake of fluorescence-labelled LDL in cells treated with CF is mediated by LDLR or CD36 (experiments showing in Fig 6). Silencing one receptor using siRNA will define the role of CD36/LDLr in regulating LDL uptake.

6- Figure 6F, description of the groups missing

Reviewer #2:

Remarks to the Author:

Proprotein Convertase Subtilisin/Kexin type 9 inhibitors (PCSK9-I) have been reported to cause a moderate increase in high-density lipoprotein (HDL) cholesterol in human studies. Currently, some PCSK9 inhibitors were available, thus, I strongly recommend that the authors use some representative inhibitors as positive agents in this study.

The chemical structures of small molecules endowed with PCSK9 inhibitory activity should be provided in the background or in supporting information section. And some closely related references should be cited, as listed below.

J Med Chem. 2019 Jul 11;62(13):6163-6174.

J Med Chem. 2018 Jul 12;61(13):5704-5718.

Besides, I suggest that the authors add an illustration (such as the effect on the relevant cell pathways) to the conclusion to express the core content of this article.

From the perspective of medicinal chemistry, small changes in the structure of small molecular compounds often have a significant impact on their biological effects. Therefore, I suggest that the authors also investigate the biological effects of caffeine analogues (to check the effects on Blocking SREBP2-induced Hepatic PCSK9 Expression to Enhance LDLR-2 Mediated Cholesterol Clearance).

Reviewer #3:

Remarks to the Author:

In this manuscript, the authors demonstrated that caffeine increased LDL uptake by blocking SREBP2-induced PCSK9 through an increased ER calcium level in various types of hepatocytes.

Caffeine-increased LDL uptake was confirmed by mice, while reduction of plasma PCSK9 level by oral given of caffeine in a human. Although the manuscript contains some novel findings, it need revision to make a better product.

Comments

1. Molecular target of caffeine is lacked in this study. As the authors mentioned in the discussion section adenosine receptors and GABA receptors are well-known molecular targets of caffeine. How caffeine works in this study? Does caffeine incorporate into hepatocytes and directly increase the ER calcium level?
2. For the cell culture study, the authors used caffeine at 200 microM. Why the authors used such high concentration of caffeine? In the animal experiments and human trial, the authors should show the plasma concentration of caffeine, and used concentration in the cell culture experiments is a relevant one. In addition, there is no dose-dependency data in the cell culture experiments.
3. Title should be reconsidered, because there is no data about cholesterol level in neither mice experiments nor human trial in current manuscript. The authors should add plasma (or serum) cholesterol level including LDL-cholesterol level in both animal and human.
4. As for expression of GRP78, the authors should confirm caffeine increased GRP78 expression and add data as a supplemental figure.
5. Part of the experiments using cultured and primary hepatocytes is well conducted and obtained results are sound. Selected important effects of caffeine should confirm in the liver of mice, because the authors only measured LDLR expression and hepatic and serum Dil-LDL fluorescence.

Response to reviewer's comments

Title: Caffeine Blocks SREBP2-Induced Hepatic PCSK9 Expression to Enhance LDLR-Mediated Cholesterol Clearance

Authors: Paul F. Lebeau^{1*}, Jae Hyun Byun^{1*}, Khrystyna Platko¹, Paul Saliba², Matthew Sguazzin², Guillaume Pare³, Gregory R. Steinberg^{2,4}, Luke J. Janssen⁵, Suleiman A. Igdoura⁶, Mark A. Tarnopolsky⁷, S. R. Wayne Chen⁸, Nabil G. Seidah⁹, Jakob Magolan² and Richard C. Austin^{1†}

General commentary from the authors in response to revisions:

It is on the behalf of all authors involved in this manuscript that we thank the editor and reviewers for providing the constructive remarks/criticisms necessary to enhance the quality of this study. In response to the comments, a variety of new experiments were conducted in order to strengthen our understanding of the mechanism by which caffeine impacts PCSK9 expression, as well as hepatic LDL uptake. Furthermore, additional control studies improve our understanding and enhance the reliability of the results presented in this revised manuscript.

Reviewer #1 (Remarks to the Author):

This study investigates the effect of caffeine in regulating cholesterol metabolism *in vitro* and *in vivo*. The authors show that CF suppress SREBP2 activation, reducing SREBP2-regulation of PCSK9 expression. While the finding is of interest, there are key experiments missing to support authors conclusion. In particular the authors should investigate the effect of CF in regulating the post-transcriptional regulation of LDLR (see below).

Reviewer comment 1- One of the major problems of the study is the lack of the experiments assessing the effect of caffeine (CF) in the posttranscriptional regulation of LDLR. Indeed, the authors did not assess the effect of CF in regulating LDLr mRNA levels. Both PCSK9 and LDLR are regulated by SREBP2, thus if CF regulates PCSK9 via SREBP2 activity and processing, LDLR mRNA should be elevated as well. However, the authors did not analyze the effect of CF on LDLR mRNA expression. Thus, additional experiments showing the post-transcriptional regulation of LDLr *in vitro* (e.g., actinomycin treatment etc) using CF are critical to support the mechanism proposed by the authors.

Author response to comment 1- We strongly agree that PCSK9 and the LDLR are regulated by SREBP2 and that data on the mRNA levels of the LDLR were lacking in the previous version of the manuscript. The updated version now includes LDLR mRNA levels in Fig. 5A and J (*in vivo*), as well as Fig. 6E (*in vitro*). Similar to PCSK9 mRNA levels, a significant reduction in LDLR mRNA levels was observed in response to caffeine. Given that caffeine also blocks SREBP2 activation and subsequent PCSK9 expression, the observed reduction in LDLR mRNA was an expected result. Interestingly, however, an increase in LDLR protein expression, including cell surface expression, is also observed and may stand as a counterintuitive outcome given the impact on mRNA levels. Importantly, as we have highlighted in the discussion of the manuscript, LDLR expression is primarily dependent on protein stability (relatively long half-life) and on factors that impact its stability (ie, PCSK9). In contrast, circulating proteins like PCSK9 are more dependent on *de novo* synthesis and expression which closely follows the activity of its transcription factor (ie, SREBP2). The text in the discussion that highlights this phenomenon is as follows: “*We also observed that CF induced the protein expression of the LDLR and increased LDLc uptake in cultured hepatocytes. Given that SREBP2 regulates de novo expression of PCSK9 and the LDLR, the observed induction of cell-surface LDLR in the face of SREBP2 inhibition likely occurs in response to the loss of circulating PCSK9 levels. Several studies have demonstrated that, with a half-life of five minutes (51), PCSK9 expression closely follows that of SREBP2 (52). In contrast, with a 144-fold increased half-life compared to PCSK9 (12 h) (53), expression of the LDLR appears less dependent on de novo synthesis and more on factors that influence its stability at the cell surface, like circulating PCSK9.*”

Reviewer comment 2- Is CF affecting PCSK9 secretion independent of the transcription?

Author response to comment 2- The authors agree with Reviewer 1 on the importance of the mechanism by which caffeine impact secreted PCSK9 levels. In the previous version of the manuscript, we had demonstrated that caffeine failed to impact the levels of secreted PCSK9 in

cells that were transfected with a plasmid that induced the expression of PCSK9 via the cytomegalovirus (CMV) transcriptional element. Given that CMV-driven PCSK9 protein was secreted to the same extent in cells treated with caffeine as in cells treated with a vehicle control, our interpretation was that caffeine does not heavily impact post-transcriptional PCSK9 processing, but rather impacts non-CMV dependent transcription of PCSK9 (ie, SREBP2 via the SRE element). To add further emphasis to this point, this figure has been moved from the supplementary file to the main manuscript as **Fig. 1L**. In addition, as the reviewer has recommended, the impact of caffeine on PCSK9 secretion was also assessed in cells treated with an inhibitor of transcription (actinomycin D). In line with Fig. 1L, results of this experiment suggest that caffeine affects PCSK9 at the transcriptional level (**Fig. 1J and K**).

Reviewer comment 3- The staining for hepatic LDLr looks non-specific (only stain the liver sinusoidal ECs). Please add a control for this staining using the LDLR null mice (Figure 5 and 6)

Author response to comment 3- The impact of caffeine on hepatic LDLR expression in response to changes in PCSK9 levels is a key finding of our study. For this reason, we agree that additional control experiments were required to support LDLR staining. We have now conducted staining in the livers of wild-type C57BL/6J mice, as well as in *Pcsk9*^{-/-} mice and *Ldlr*^{-/-} mice. These data clearly demonstrate punctate staining for the LDLR at the cell surface of liver hepatocytes in the wild-type mice, with increased expression in the *Pcsk9*^{-/-} mice (**Fig. S7**). In contrast, a complete absence of LDLR staining was observed in *Ldlr*^{-/-} mice. In further support of these data, a substantial increase in hepatic cell-surface LDLR expression was also observed in the livers of mice treated with alirocumab, an established inhibitor of PCSK9 (**Fig. 5H and I**).

Reviewer comment 4- WB for LDLr in Figure 5F should be repeated (poor quality and overexposed)

Author response to comment 4- The immunoblot previously presented in **Fig. 5F** has been replaced with a higher quality immunoblot with reduced exposure.

Reviewer comment 5- CD36 also bind and uptake native LDL. Thus, it is not clear whether the increase uptake of fluorescence-labelled LDL in cells treated with CF is mediated by LDLR or CD36 (experiments showing in Fig 6). Silencing one receptor using siRNA will define the role of CD36/LDLr in regulating LDL uptake.

Author response to comment 5- We agree with Reviewer 1 that LDL has been shown to bind CD36 in certain instances (albeit rare). Therefore, we have conducted additional experiments to characterize any potential role that CD36 may have in LDLR uptake in response to caffeine. In separate experiments, CD36 expression was blocked using siRNA as well as a pharmacologic inhibitor (SSO). In both instances, inhibition of CD36 failed to impact caffeine-mediated LDL uptake, suggesting that CD36 does not play a major role in caffeine-mediated LDL uptake in hepatocytes (**Fig. 6F and H**). Effective knockdown of CD36 was confirmed via immunoblot (**Fig. 6G**).

Reviewer comment 6- Figure 6F, description of the groups missing

Author response to comment 6- A description of the groups has been added.

Reviewer #2 (Remarks to the Author):

Proprotein Convertase Subtilisin/Kexin type 9 inhibitors (PCSK9-I) have been reported to cause a moderate increase in high-density lipoprotein (HDL) cholesterol in human studies.

Reviewer comment 1 - Currently, some PCSK9 inhibitors were available, thus, I strongly recommend that the authors use some representative inhibitors as positive agents in this study.

Author response 1- We agree with Reviewer 2 that implementing established PCSK9 inhibitors as controls in our study would add perspective to the results and overarching message of the manuscript. As such, we have treated male C57BL/6J mice with alirocumab (anti-PCSK9 mAb) for 10 days and assessed hepatic LDLR expression via immunohistochemical staining, immunoblotting and RT-PCR. Results from this experiment support the notion that inhibitors of PCSK9 yield elevated LDLR expression levels in the liver.

Reviewer comment 2- The chemical structures of small molecules endowed with PCSK9 inhibitory activity should be provided in the background or in supporting information section. And some closely related references should be cited, as listed below.
J Med Chem. 2019 Jul 11;62(13):6163-6174.
J Med Chem. 2018 Jul 12;61(13):5704-5718.

Author response 2- We thank Reviewer 2 for bringing these very interesting studies to our attention. We have now included these references in the manuscript and provided a comparison with caffeine. We have also include the structures of the novel PCSK9 inhibitors (MLRA-1812/1820); supplemental Table 2.

Reviewer comment 3- Besides, I suggest that the authors add an illustration (such as the effect on the relevant cell pathways) to the conclusion to express the core content of this article.

Author response 3- We agree with Reviewer 2 and have added a graphical abstract as figure 9 of the manuscript to increase the clarity of the pathways involved.

Reviewer comment 4- From the perspective of medicinal chemistry, small changes in the structure of small molecular compounds often have a significant impact on their biological effects. Therefore, I suggest that the authors also investigate the biological effects of caffeine analogues (to check the effects on Blocking SREBP2-induced Hepatic PCSK9 Expression to Enhance LDLR-2 Mediated Cholesterol Clearance).

Author response 4- The authors agree with Reviewer 2 and have conducted several experiments to further characterize the impact of the caffeine derivatives on LDLR expression and LDL uptake. In response to treatment with MLRA-1812 and MLRA-1820, we observed a significant increase in LDLR expression using immunofluorescence microscopy (staining was also

quantified using ImageJ software; **Fig. 8F**). Exposure of these cells to DiI-LDL also demonstrated that the novel derivatives significantly increased LDL uptake (**Fig. 8G**).

Reviewer #3 (Remarks to the Author):

In this manuscript, the authors demonstrated that caffeine increased LDL uptake by blocking SREBP2-induced PCSK9 through an increased ER calcium level in various types of hepatocytes. Caffeine-increased LDL uptake was confirmed by mice, while reduction of plasma PCSK9 level by oral given of caffeine in a human. Although the manuscript contains some novel findings, it need revision to make a better product.

Reviewer comment 1- Molecular target of caffeine is lacked in this study. As the authors mentioned in the discussion section adenosine receptors and GABA receptors are well-known molecular targets of caffeine. How caffeine works in this study? Does caffeine incorporate into hepatocytes and directly increase the ER calcium level?

Author response -1 In this study, we characterize the ability of caffeine to increase ER Ca²⁺ levels as the mechanism by which caffeine impacts SREBP2 activation and subsequent PCSK9 inhibition. In future studies, our research group aims to characterize *how* caffeine impacts ER Ca²⁺ levels in hepatocytes. Although this was not directly within the scope of the present study, we have added additional content in the discussion on the potential targets by which caffeine may be affecting ER Ca²⁺ levels: *“CF is known to exert its effect through a range of molecular targets, including the antagonism of adenosine receptors, GABA receptors and phosphodiesterase enzymes, as well as inducing intracellular Ca²⁺ transients by enhancing RyR-mediated calcium-induced calcium release (CICR) (11). Although the aforementioned interactions do not directly support our observation, in which CF increases ER Ca²⁺ levels, CF is also known to block ER Ca²⁺ release via inhibition of the IP3-receptor (36, 37). Given the broad range of targets known to interact with CF, the identification of exact molecular mechanisms pertaining to its protective effect on the vascular system is challenging.”* Our preliminary data suggest that the process by which caffeine impacts ER Ca²⁺ levels depends on the classical ER Ca²⁺ channels and pumps (eg, IP3R, RyR and SERCA).

Reviewer comment 2- For the cell culture study, the authors used caffeine at 200 microM. Why the authors used such high concentration of caffeine? In the animal experiments and human trial, the authors should show the plasma concentration of caffeine, and used concentration in the cell culture experiments is a relevant one. In addition, there is no dose-dependency data in the cell culture experiments.

Author response 2- We agree with Reviewer 3 that 200 μM can be considered a fairly high concentration. For this reason, and based on the recommendation by the reviewer, we have now added a dose-response experiment (**Fig.1 I**) to the study results. These data demonstrate that hepatic PCSK9 expression is impacted by caffeine at increasing doses of 100 nM and upwards.

Reviewer comment 3- Title should be reconsidered, because there is no data about cholesterol level in neither mice experiments nor human trial in current manuscript. The authors should add plasma (or serum) cholesterol level including LDL-cholesterol level in both animal and human.

Author response 3- In the original version of the manuscript, we had demonstrated that caffeine significantly increased the uptake of DiI-LDL cholesterol into the liver, while reducing the levels of DiI-LDL cholesterol in the circulation, following a bolus injection in mice (Currently **Fig. 6J and K**). We conducted additional experiments, in which mice were treated with caffeine on a daily basis for two weeks. Circulating levels of ApoB, a surrogate marker of native LDLc, were then quantified using ELISAs. A significant reduction in circulating ApoB and PCSK9 levels were observed in response to daily caffeine treatment (**Fig. 6L and M**). Typically, circulating LDLc levels require a minimum of two weeks to be impacted by new treatments and so, assessment of circulating LDLc was not included as an outcome in our approved clinical trial protocol. Assessment of LDLc, however, will be included as an endpoint in future clinical studies, in which volunteers will be treated with caffeine for longer time periods. Overall, we believe that the data added to the revised version of the manuscript now support the title.

Reviewer comment 4- As for expression of GRP78, the authors should confirm caffeine *increased* GRP78 expression and add data as a supplemental figure.

Author Response 4- We have observed that treatment with caffeine leads to a net reduction in the expression of several ER stress markers/UPR chaperones, including GRP78. Our data demonstrate that caffeine reduces the abundance of GRP78 (**Fig. 4G and 5F**), but increases the binding of GRP78 for ER-resident SREBP2 (**Fig. 4A**). This notion is supported by previous studies demonstrating that Ca^{2+} is necessary for the chaperone functionality of GRP78 (PMID: 16418174; PMID: 33295873). Dissociation of GRP78 from the SREBP complex has also been characterized as a mechanism leading to the activation of SREBP (PMID:19363290).

Reviewer comment 5- Part of the experiments using cultured and primary hepatocytes is well conducted and obtained results are sound. Selected important effects of caffeine should confirm in the liver of mice, because the authors only measured LDLR expression and hepatic and serum DiI-LDL fluorescence.

Author response 5- We agree with Reviewer 3 and have added additional data to the manuscript. We now demonstrate that caffeine (i) blocks the mRNA expression of PCSK9 in the livers of mice (**Fig. 5G**), (ii) reduces the protein abundance of PCSK9 in the serum of mice (**Fig. 5A and C**), (iii) increases the protein abundance of the LDLR in the livers of mice (**Fig. 5D, 5F, 6I**), and (iv) increases the uptake of LDL cholesterol by the liver while reducing circulating levels of LDLc (**Fig. 6J, 6K and 6L**).

Reviewers' Comments:

Reviewer #1:

Remarks to the Author:

The authors have addressed my previous concerns

Reviewer #2:

None

Reviewer #3:

Remarks to the Author:

Thank you for your response against the comments. This reviewer confirmed that the authors well addressed to the comments and added the data. Thus, this reviewer has no more comment.

Response to reviewer's comments (2021-11-22)

Title: Caffeine Blocks SREBP2-Induced Hepatic PCSK9 Expression to Enhance LDLR-Mediated Cholesterol Clearance

Authors: Paul F. Lebeau^{1*}, Jae Hyun Byun^{1*}, Khrystyna Platko¹, Paul Saliba², Matthew Sguazzin², Guillaume Pare³, Gregory R. Steinberg^{2,4}, Luke J. Janssen⁵, Suleiman A. Igdoura⁶, Mark A. Tarnopolsky⁷, S. R. Wayne Chen⁸, Nabil G. Seidah⁹, Jakob Magolan² and Richard C. Austin^{1†}

General commentary from the authors in response to revisions:

We would like to thank the reviewer's and editorial team once again for the continued support with the assistance on this manuscript. Although no changes were requested by the reviewers in this revision, many changes were made to the formatting of figures, as well as a variety of in-text changes based on the formatting requirements of Nature Communications. We hope to have addressed all of the concerns, however, please do not hesitate to contact our group again if other questions/queries should arise.